# Pentavalent and tetravalent uranium formation via glycerol-stimulated bacteria in mine water

Antonio M. Newman-Portela [1,2] ✉, Kristina O. Kvashnina [1,3] ✉,
Elena F. Bazarkina[1,3,5], André Rossberg[1,3], Frank Bok [1], Sean Ting-Shyang Wei [1],
Andrea Kassahun[4], Thorsten Stumpf[1], Johannes Raff [1],
Mohamed L. Merroun[2] ✉ & Evelyn Krawczyk-Bärsch [1] ✉

Uranium contamination is a major global concern, as its chemical and radiological effects threaten ecosystems and human health, particularly in mining-impacted regions. Pentavalent uranium is a key but often overlooked intermediate in U biogeochemistry, challenging the conventional view of a direct U(VI) to U(IV) transition and precipitation in natural waters. Here we demonstrate the formation and stability of U(V) under environmentally relevant mine-water conditions using advanced spectroscopic and microscopic analyses. Our data reveal concurrent U(VI) reduction to U(IV), as biogenic uraninite nanoparticles, and to U(V) as $FeU^{(V)}O_4$ nanoparticles and U(V)-carbonate complexes. U(V) persists for at least 130 days under anoxic conditions and four weeks after exposure to oxygen. Microbial community analysis reveals enrichment of fermentative and sulphate-reducing taxa, consistent with redox conditions favouring U reduction. By demonstrating the stability of immobilised U(V) alongside U(IV), this work advances the understanding of uranium biogeochemistry and offers new insights for sustainable remediation strategies.

Globally, uranium (U) contamination remains a major environmental challenge, particularly in regions impacted by historical and active U mining activities[1]. Additionally, other anthropogenic sources, such as phosphate fertilisers, have further contributed to its widespread occurrence[2,3]. Elevated U concentrations in surface and groundwater frequently exceed the drinking-water guideline of 0.03 mg mL$^{-1}$[4]. Such levels have been reported in many countries including the United States, India, Canada, France, South Africa and Australia, underscoring the magnitude of the problem[1,2]. Similar cases of long-term U contamination have also been reported in Brazil and Portugal, illustrating the global persistence of mining-related U pollution[1]. In Europe, Germany is one of several countries addressing this mining legacy, with

the Schlema-Alberoda site in Saxony, located in the Ore Mountains, serving as a representative case study. This site was among the world's largest U mining operations until its closure in 1990. Since then, the underground mine has been flooded and the resulting mine water requires continuous treatment[5].

U occurs in mine water mainly as hexavalent uranium (U(VI)) under oxidising conditions, typically as the uranyl ion ($UO_2^{2+}$) and its aqueous complexes with carbonates[6,7]. Upon reduction to tetravalent uranium (U(IV)), it becomes less mobile and most commonly precipitates as uraninite ($U^{IV}O_2$)[2,8]. Physico-chemical mine water treatment applied at Schlema-Alberoda has proven to be effective in substantially reducing contaminant concentrations in the discharge water[5,9,10]. The

[1]Institute of Resource Ecology, Helmholtz-Zentrum Dresden-Rossendorf, Dresden, Germany. [2]Faculty of Science, Department of Microbiology, University of Granada, Granada, Spain. [3]The Rossendorf Beamline (BM20-ROBL), European Synchrotron Radiation Facility, Grenoble, France. [4]WISMUT GmbH, Chemnitz, Germany. [5]Present address: Institut des Sciences de la Terre (ISTerre), Grenoble, France. ✉e-mail: a.newman-portela@hzdr.de; kristina.kvashnina@esrf.fr; merroun@ugr.es; e.krawczyk-baersch@hzdr.de

approach relies on a modified lime precipitation process, in which alkaline reagents are added to promote the precipitation of radionuclides and other contaminants, after aeration and $CO_2$-stripping. The resulting precipitates are then removed by solid-liquid separation, followed by sludge management[5,9,10]. During three decades after initial mine flooding, dissolved U concentrations have decreased by nearly one order of magnitude. However, mine water still contains ~1 mg mL$^{-1}$ U(VI), well above the discharge limits applicable in Saxony (0.20−0.50 mg mL$^{-1}$)[5]. The applied on-site mine water treatment requires continuous operation, generates large volumes of secondary waste and remains costly, with no foreseeable endpoint. Complementary strategies to accelerate contaminant concentrations decline within the flooded mine could help shorten the duration of active mine water treatment.

The untreated mine water represents risks to human health and the environment. Such persistent contamination also prevents mine water from meeting the standards required for water reuse. Over the last three decades, bioremediation has been explored as a cost-effective alternative to physico-chemical water treatment[2]. Field studies have demonstrated substantial U reduction while avoiding the generation of secondary sludge, with treatment performance closely influenced by site-specific redox dynamics and geochemical conditions[11,12]. U bioremediation relies mainly on two microbial processes, non-reductive biomineralisation and bioreduction[2,13]. Non-reductive biomineralisation involves the precipitation of U without changing its oxidation state, most commonly through the formation of stable phosphate minerals[2,14]. Bioreduction occurs when microorganisms enzymatically reduce soluble U(VI) to U(IV), resulting in the precipitation of insoluble U(IV) phases, such as uraninite, via microbial electron transfer[2,6,15]. This process is mediated by diverse anaerobic bacteria, including metal-reducing taxa (*Geobacter*, *Shewanella*), sulphate-reducing taxa (*Desulfovibrio*, *Desulfosporosinus*), and fermentative and denitrifying microorganisms (*Clostridium* and *Pseudomonas*)[2,6,13,15–18]. External electron donors such as acetate, lactate, or hydrogen enhance bioreduction by stimulating indigenous U(VI)-reducing microbes[3,18,19]. While electron donors can be cost-effective at the laboratory scale, their sustained supply in field applications often poses considerable economic challenges. In contrast, biodiesel production generates large volumes of crude glycerol, a low-value impure byproduct[20]. As result, crude glycerol is abundant, regionally available and low-cost electron donor, offering a practical and economically viable option for U bioremediation at larger scales. In our system, glycerol outperformed other tested electron donors (vanillic acid and gluconic acid), highlighting its potential under site-specific conditions[3].

Among U oxidation states, the pentavalent U (U(V)) represents a transient redox state with major implications for U mobility in contaminated environments[21,22]. It has been detected as $UO_2^+$ ion in abiotic and biotic reduction experiments, in synthetic compounds and in the mineral wyartite, the first known U(V) mineral[15–17,23–28]. Although generally considered unstable, U(V) can persist under specific conditions[21,29,30]. Its stabilisation has been reported through carbonate complexation, coordination with aminocarboxylate ligands and incorporation into Fe-bearing (oxyhydr)oxides such as magnetite and green rust[21,23,24]. FeU$^{(V)}O_4$ identified in depleted U residues has remained stable for more than 25 years, while biogenic U(V) phases have been observed for over 120 h under anoxic conditions[17,31]. U(V) has also been detected during microbial U(VI) reduction by iron-reducing bacteria (IRB; *Geobacter sulfurreducens*, *Shewanella oneidensis*) and sulphate-reducing bacteria (SRB; *Desulfosporosinus hippei*)[15–17]. To date, however, U(V) formation has only been reported in pure bacterial cultures under millimolar U(VI) and chemically simplified conditions[16,17]. Such conditions do not represent mine waters, which typically contain micromolar U and host diverse microbial communities[2,3,7,32]. It is crucial to determine whether U(V) can form and persist under environmentally relevant conditions, as these factors ultimately govern U mobility and the long-term effectiveness of remediation strategies.

In this study, a key question we addressed was whether glycerol-stimulated microbial communities can generate U reduction products that remain stable under environmentally relevant mine-water conditions. Using advanced spectroscopic, microscopic and geochemical techniques, we aimed to identify and characterise the U(VI) reduction products, including U(IV) and U(V), and to assess their stability. The key U species observed were U(IV) as biogenic uraninite ($UO_2$), FeU$^{(V)}O_4$ nanoparticles (NPs) and U(V)-carbonate complexes. These results demonstrate the environmental relevance of U(V) as a stable reduced species and highlight its role as a stabilising intermediate in U biogeochemistry, with direct implications for long-term remediation strategies.

## Results and discussion

### Geochemistry and kinetics of U reduction in microcosms

Glycerol-amended mine water microcosms were incubated for 130 days under anoxic conditions. This setup aimed to evaluate the ability of indigenous U-reducing microorganisms to decrease U concentrations from an initial value of 1 mg mL$^{-1}$ in Schlema-Alberoda mine water (Supplementary Fig. 1). The monitoring of the physico-chemical parameters (pH, $E_h$, U, Fe, As and $SO_4^{2-}$) showed a nearly constant pH of 7.5–8.0, but a remarkable downward trend in the $E_h$ values, ranging from +398 at the beginning to −114 mV at the end of the experiment (Supplementary Fig. 2). The reducing conditions, triggered by the biostimulation of the microbial community, resulted in a remarkable decrease of the U(VI) concentration up to 96% from 1 to 0.04 mg mL$^{-1}$ (Supplementary Fig. 3). During the first 20 days of the experiment, the decrease was modest, 5–20%, but dropped notably after 20 days (up to 96%) (Supplementary Table 1). Slight changes in U concentrations were also observed in control microcosms. After 130 days, U(VI) decreased by approximately 25% in the glycerol-free control and by 36% in the glycerol-containing autoclaved control (Supplementary Table 2). In both cases, the decrease in U concentration is best explained by abiotic or adsorptive removal, which may include adsorption onto biomass in the glycerol-free control, as well as adsorption on container surfaces, or mineral particulates present in the mine water in both controls[6,22,33,34]. In the glycerol-enriched microcosm, the concentrations of Fe, $SO_4^{2-}$ and As also decreased considerably by about 98%, 68% and 44%, respectively, at the end of the experiment. Fe and $SO_4^{2-}$ are terminal electron acceptors that have a higher thermodynamic preference for microbes than U under standard conditions[8]. The decreased amounts of Fe and $SO_4^{2-}$ could be mainly attributed to the activity of SRB, particularly of the genera *Desulfobulbus* and *Desulfovibrio*. These groups were enriched in the sample studied and will be analysed in more detail below. Both are closely associated with U(VI) reduction, by using U(VI) as an electron acceptor[8,35]. However, the possible indirect abiotic reduction of U by biogenic hydrogen sulphide and/or Fe(II) cannot be excluded[36,37].

The measured parameters, including pH and $E_h$, obtained during the experiment at different incubation times (up to 130 days), were used to calculate the predominance fields of potential U species. The pourbaix diagram (Supplementary Fig. 4), calculated with the geochemical speciation code Geochemist's Workbench (v18.0.3)[38] and the PSI Chemical Thermodynamic Database 2020[39], predicts an aqueous $Ca_2UO_2(CO_3)_3$ species as the dominant species at the beginning of the experiment. Previous spectroscopic studies even identified the two U species, $UO_2(CO_3)_3^{4-}$, and $Ca_2UO_2(CO_3)_3(aq)$[3]. Mono-, di- and tri-uranyl carbonate species are generally dominant at neutral to alkaline pH[40]. Interestingly, the presence of $Ca_2UO_2(CO_3)_3$ in U-contaminated groundwater has been reported to result in incomplete removal of soluble U[41], whereas we observed almost complete removal. The reduction of U(VI) to U(IV) is predicted due to the biostimulation of the

indigenous bacteria and the resulting decrease in $E_h$. This indicates the potential formation of the solid U(IV) species, uraninite, as the main reduction product after 40–52 days.

## Spectroscopic and microscopic analysis of bioreduced U(VI)

To enable a consistent comparison of U speciation during the bioreduction process, spectroscopic and microscopic analyses were performed on black precipitates collected at three defined sampling points, selected to represent distinct chemical and redox states of the microcosms. These sampling points correspond to conditions at which the dissolved U(VI) concentration in the aqueous supernatant had decreased by approximately 20%, 60% and 90% relative to the initial value.

Based on the temporal evolution of dissolved U(VI) concentrations and redox potential (Supplementary Fig. 2), these sampling points were reached after approximately 10 days (day 10), 30 days (days 20–40) and 55 days (days 52–60) of incubation, respectively. Because U reduction in these systems is biologically mediated and progresses continuously over time, the correspondence between incubation time and U removal is inherently approximate. The reported percentages therefore serve as operational descriptors of the system state at the time of sampling, rather than defining discrete kinetic stages.

A combination of U $M_4$-edge high-energy-resolution fluorescence-detected X-ray absorption near-edge structure (HERFD-XANES) spectroscopy and iterative target factor analysis (ITFA) was used to determine the oxidation state of U in the black precipitates formed in different Schlema-Alberoda mine water microcosms amended with glycerol. Figure 1 shows the HERFD-XANES spectra and a bar plot showing the relative percentage of different U oxidation states in black precipitates collected at the three defined sampling points. In the precipitate collected after ~10 days of incubation (~20% decrease in dissolved U(VI) in the supernatant), the spectrum indicates the presence of U(VI) (30%), characterised by a feature at ~3726.5 eV, together with U(IV) (50%) at ~3725.0 eV. A similar pattern is observed in the black precipitate collected after ~30 days (~60% decrease), with U(VI) (20%) being less pronounced and U(IV) (60%) being more pronounced. In the precipitate collected after ~55 days (~90% decrease), the spectrum of the black precipitate shows no U(VI) features, with U(IV) (70%) being the dominant oxidation state, along with a shoulder at ~3726.0 eV indicating the presence of U(V) (30%). The results showed that U(IV) was the dominant oxidation state in all three analysed black precipitate samples. Additionally, the proportions of U(VI) in the black precipitates decreased as U concentrations in the supernatants decreased. These findings are consistent with other studies that report U(IV) as the dominant oxidation state during microbial U(VI) reduction by pure bacterial cultures[16,17]. The outstanding finding was the presence of U(V) in all samples with high proportions ranging from 20% to 30%. This indicates that microbial reduction of U(VI) in the microcosm experiments is not limited to U(IV), but may also include U(V) phases.

While HERFD-XANES provided high-resolution detection of oxidation states, U $L_3$-edge extended X-ray absorption fine structure (EXAFS) provided deeper insights into U speciation and local structure. Their combination allowed a comprehensive characterisation of biogenic U phases, distinguishing coordination environments and potential U(V) species. U $L_3$-edge $k^3$-weighted EXAFS spectra and their corresponding phase shift uncorrected Fourier transforms (FT) reveal systematic differences in U speciation across the black precipitate samples collected at the defined sampling points (Supplementary Fig. 5). The FT peak at $1.43 + \Delta\text{Å}$ (Supplementary Fig. 5, peak 1), which is characteristic of uranyl axial oxygen ($O_{ax}$), decreases with decreasing U concentration in the supernatant. Furthermore, the peak at $3.64 + \Delta\text{Å}$ (Supplementary Fig. 5, peak 2), which originates from a U–U interaction, exhibits a simultaneous progressive increase in amplitude. Thus, structurally distinct U species coexist, causing the observed changes in

the FT peak amplitudes as their relative proportions in their spectral mixtures vary. ITFA estimates two distinct U species ($n = 2$), such that all the spectral mixtures can be reproduced by linear combinations of two spectral components (Supplementary Fig. 5). To identify the mixed chemical species, we applied target factor analysis (TFA)[16,42] using a database of 81 EXAFS reference spectra from inorganic and organic U-compounds in oxidation states U(IV, V, VI). The best-fitting references identified by TFA are uraninite phases (U(IV)), solid and aqueous U(VI)- and U(V)-carbonates such as $Na_4UO_2(CO_3)_3$, $UO_2(CO_3)_3^{4-}$ and electrochemically synthesised $UO_2(CO_3)_3^{5-}$, respectively (Fig. 2). Note that the latter species contain partially $UO_2(CO_3)_3^{4-}$, hence U(VI) which was not completely reduced to U(V). Before fitting of pure U species, their spectra were isolated from the spectral mixtures. Following the isolation procedure described by Hilpmann et al.[16], we assumed one U species as the pure uraninite-like phase and minimised the U–U interaction peak ($3.64 + \Delta\text{Å}$, Supplementary Fig. 5, FT peak 2) in the other. In the ITFA-isolated spectrum (a) (Supplementary Fig. 6), the U signal at $3.64 + \Delta\text{Å}$ is minimised, indicating no U(IV) contamination, while the spectrum (b) (Supplementary Fig. 6) is consistent with the measured spectrum (Supplementary Fig. 5c). Figure 2 shows the EXAFS shell fitting results for both isolated spectra and the best-fitting U references, while the corresponding structural parameters are listed in Table 1.

The amplitude reduction factors ($S_0^2$) were estimated by assuming an average coordination number (CN) of the first shell oxygen atoms of CN = 2 and CN = 8 for the carbonato references and the uraninite species, respectively. For the isolated signal of the U(V/VI) species, $S_0^2$ was fixed at 0.96, as the derived EXAFS structural parameters closely resemble those obtained for the U–carbonato reference compounds. Compared to the U–carbonato references, a stronger correlation between the CN of the $O_{ax}$ and $O_{eq}$ shells was observed. To avoid these correlation effects, the coordination number of $O_{ax}$ was fixed at CN = 2, while CN = 5 was assumed for $O_{eq}$, reflecting the commonly observed four- to six-fold equatorial coordination of U(V) and U(VI)[43]. However, the determination of CN might carries high uncertainty, hence limiting reliable conclusions, the analysis of radial distances (R) is more conclusive due to its low absolute error in determination ($\pm0.02$ Å)[44] except for the fitted carbon shell with a higher uncertainty of $\pm0.04$ Å. The U(V/VI) species exhibits a high $R_{Oax} = 1.85$ Å, aligning more closely with U(V) ($UO_2(CO_3)_3^{5-}$: 1.86 Å) than with U(VI) species ($Na_4UO_2(CO_3)_3$: 1.82 Å, $UO_2(CO_3)_3^{4-}$: 1.80 Å). Given the reducing conditions in the microcosms, stable aqueous U(VI) complexes are not expected. As $R_{Oax}$ in U(V) compounds is generally higher than in U(VI) counterparts, U(V) is likely at high supernatant U concentrations (Supplementary Fig. 5a), though U(VI) cannot be excluded based on TFA references. All identified species interact with carbonate groups, suggesting a mixture of uranyl-carbonate complexes in the U(V) and U(VI) oxidation states. The relatively high DW obtained for the $O_{ax}$, $O_{eq}$ and C shells (0.013 Å$^2$, 0.008 Å$^2$ and 0.014 Å$^2$, respectively) indicate increased structural disorder and support this interpretation. Additional evidence comes from the relatively long $R_{Oeq}$ distance (2.46 Å), aligning with aqueous $UO_2(CO_3)_3^{4-}$ and $UO_2(CO_3)_3^{5-}$ (2.44–2.48 Å), in which carbonate groups are bidentately coordinated to U(V) and U(VI). This bidentate binding mode is further supported by FT features in the 2.84–4.12 Å range (Fig. 2), corresponding to strong MS paths with linearly arranged C and distal O atoms (U-C-$O_{dist}$)[45]. The isolated uranyl species lacks these MS FT features, likely masked by experimental error (Fig. 2). However, the relatively long $R_{Oax}$ (1.85 Å) and $R_{Oeq}$ (2.46 Å), along with fitted C atoms at 2.93 Å consistent with references at $R_C = 2.88$–2.94 Å, strongly indicate the presence of bidentate carbonate groups bound to U(VI) and U(V). For the uraninite-like species, $R_O = 2.33$ Å and $R_U = 3.84$ Å align within the $\pm0.02$ Å error range with uraninite references ($R_O = 2.32$ Å, $R_U = 3.84$ Å), which supports the conclusion that the biogenic species corresponds structurally to uraninite at low U concentrations in the microcosms.

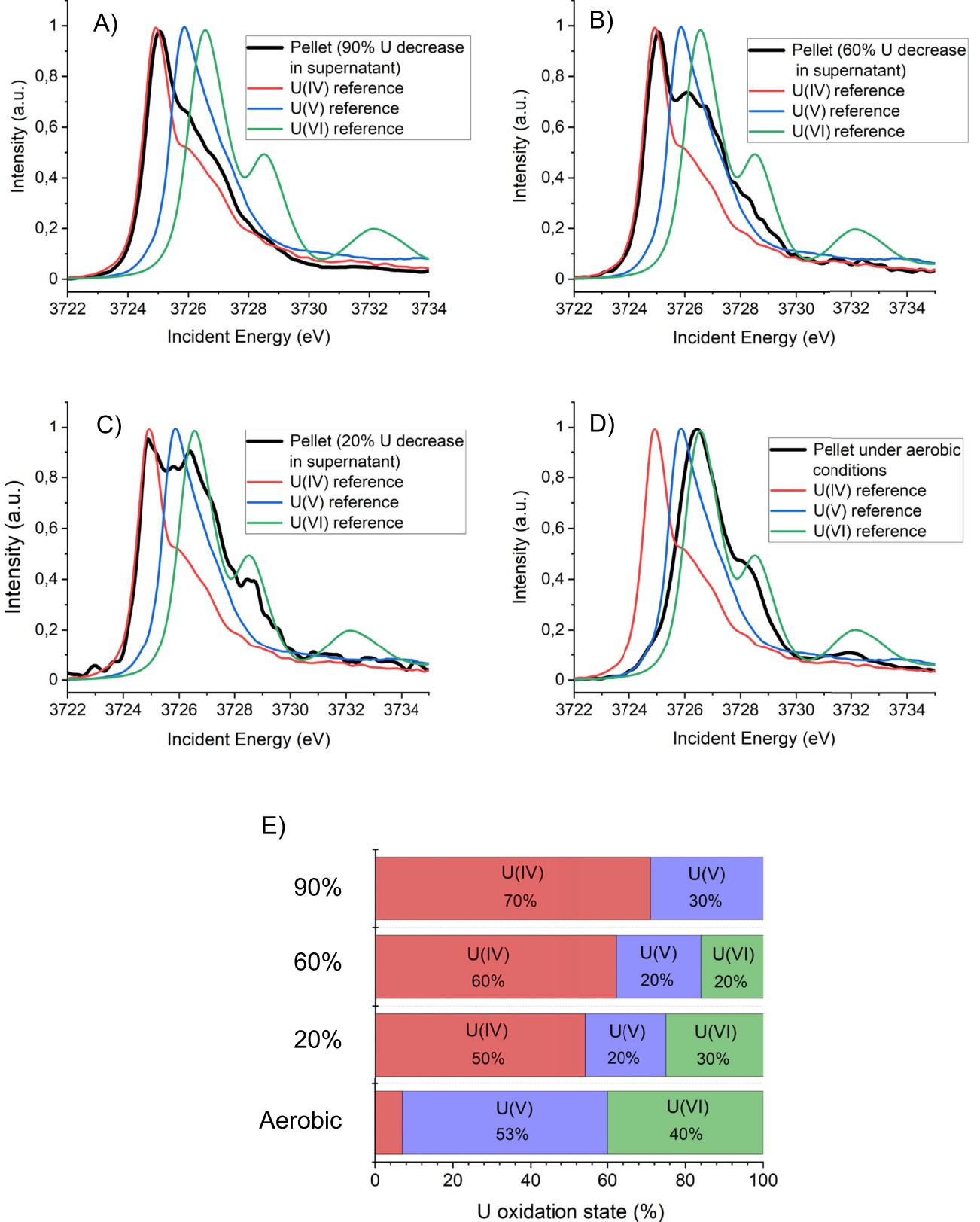

**Fig. 1 | HERFD-XANES identification of uranium oxidation states during U reduction. A–C** HERFD-XANES spectra at the U M₄-edge, recorded from the black precipitates collected at three defined sampling points, corresponding to conditions at which the dissolved U(VI) concentration in the aqueous supernatant had decreased by approximately 20%, 60% and 90% relative to the initial value. These sampling points were reached after approximately 10 days (day 10), 30 days (days 20–40) and 55 days (days 52–60) of incubation, respectively (see Supplementary

Fig. 2). Spectra are compared with U reference compounds: U(IV) as $UO_2$, U(V) as $UMoO_5$ and U(VI) as $UO_2^{2+}$. **D** HERFD-XANES spectra at the U M₄-edge, recorded from the sample exposed to oxidising conditions during 4 weeks. **E** Corresponding fractions of U(VI), U(V) and U(IV) as determined by ITFA analysis (see Supplementary Methods 1). Fractions were renormalised to sum exactly to 100%. Estimated uncertainties are ±5% for U(IV) and ±10% for U(V)/U(VI).

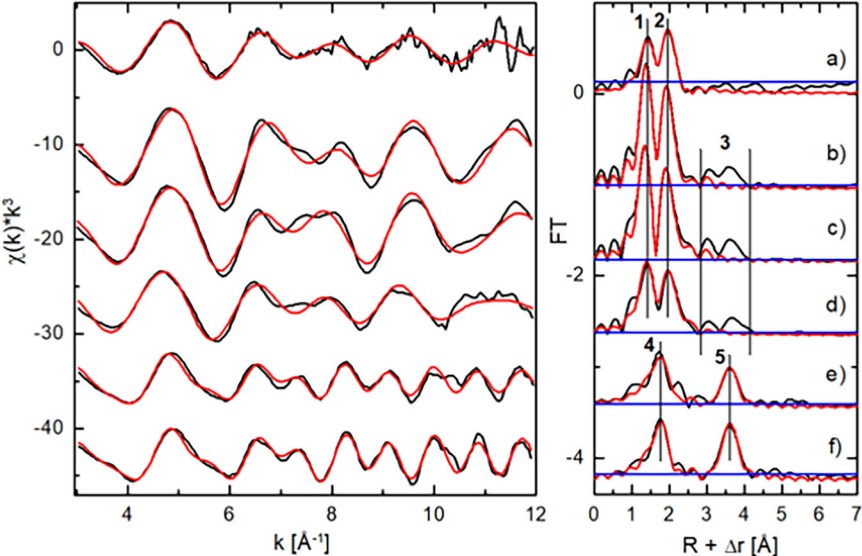

**Fig. 2 | U L₃-edge EXAFS spectra and Fourier transforms of isolated uranium species.** U L₃-edge $k^3$-weighted EXAFS spectra (left) and corresponding Fourier-transforms (FT, right) of the ITFA isolated U species (**a**, **e**) and the best-fitting reference compounds $Na_4UO_2(CO_3)_3$ (**b**), $UO_2(CO_3)_3^{4-}$ (**c**), $UO_2(CO_3)_3^{5-}$ (**d**), and colloidal uraninite (**f**) (black) with shell fit (red). Peaks are axial O atoms ($O_{ax}$) (1), equatorial O atoms ($O_{eq}$) (2), multiple scattering (MS) paths including distal O atoms ($O_{dist}$) related with U-$O_{eq}$-$O_{ax}$ and U-C-$O_{dist}$ in the R-range (3) (not fitted), O coordinated to U(IV) (4), U-U interaction (5). Level of experimental error conservatively estimated as the FT maximum in the range 6–25 Å (blue line).

Furthermore, high-angle annular dark-field scanning transmission electron microscopy (HAADF-STEM) coupled with energy-dispersive X-ray spectroscopy (EDXS) was used to investigate the cellular location, structure and crystallography properties of the U(VI) bioreduction products. To assess whether nanoparticle formation was consistent across cells, panoramic electron microscopy images were obtained and are shown in Supplementary Fig. 7. These images display multiple complete cells within the same field of view (Supplementary Fig. 7A, B) and reveal that the electron-dense agglomerates appear as bright features with a reproducible cellular distribution across independent cells (Supplementary Fig. 7C–F). In Fig. 3A–B, HAADF-STEM micrographs of a thin section of a bacterial cell from the black precipitate collected at the sampling point corresponding to ~90% decrease in dissolved U(VI) in the supernatant (reached after ~55 days of incubation) show electron-dense agglomerates on the cell surface. A distinct distribution of elements was observed, with Fe and S mainly located in the inner part, while U was predominantly found throughout the entire agglomerate (Fig. 3E–H). HAADF-STEM, together with selected area electron diffraction (SAED) and high-resolution transmission electron microscopy (HRTEM), provided evidence for few-nanometre nanoparticles within the agglomerates (Fig. 4D–E). NPs with d-spacings of 0.273 and 0.315 nm correspond to the crystallographic planes of uraninite {111} and {200} based on the American Mineralogist Crystal Structure Database. Although no clear evidence of crystallinity was observed via SAED, it is highly likely that the nanoscale size of these structures justifies this lack of evidence. The biogenic formation of uraninite is commonly reported in the literature[41,46,47]. However, other authors have reported the formation of different types of U(IV) species, such as ningyoite and coffinite, in U(VI) reduction processes by environmental bacteria[18,41,48]. This finding is particularly important as it demonstrates the microbial reduction of U(VI) to well-defined uraninite nanocrystals under environmentally relevant conditions. Unlike amorphous or weakly crystalline U(IV) phases, the identification of nanoscale uraninite with distinct crystallographic planes. This suggests a structured formation process, possibly influenced by microbial interactions and local geochemical conditions. These results contribute to the growing body of evidence that microbial communities can facilitate the formation of nanoscale crystalline U phases, leading to the immobilisation of U in contaminated environments.

However, an outstanding result is the identification of NPs with d-spacings of 0.308 nm, 0.338 nm primarily and 0.298 nm to a lesser extent, which have been correlated with $FeU^{(V)}O_4$, according to the American Mineralogist database[49]. The presence of stable U(V) in the $FeU^{(V)}O_4$ structure has been confirmed, although it is true that various possible combinations can be assumed based on the oxidation states of Fe and U, potentially incorporating U(VI), U(V), or U(IV) in their structure[50]. However, the crystal structure of $FeU^{(V)}O_4$, with U(V)-Fe(III) pairing is substantially more thermodynamically stable than the U(VI)-Fe(II) pairing[51]. Powder X-ray diffraction results indicate that the microbial reduction of U(VI) produces both $UO_2$ and $FeUO_4$, the latter being designated U(VI) in its oxidation state[52]. The HERFD-XANES results and the reducing microcosm conditions with high U(V) proportions suggest pentavalent U. However, as HERFD-XANES alone cannot distinguish U(V) species, the identification of $FeU^{(V)}O_4$ is mainly supported by structural and crystallographic analyses, with HRTEM providing crucial evidence. The combined use of these techniques provides strong evidence for the presence of $FeU^{(V)}O_4$ as a stable U(V) phase in our system, originating directly and/or indirectly from biotic processes. Pyrite NPs were also identified, likely due to SRB-driven sulphide production and microbial Fe(III) reduction. This is supported by reduced $SO_4^{2-}$ and Fe concentrations in geochemical analyses, as well as by d-spacing data and comparison with the American Mineralogist database[53]. However, pyrite NPs were less abundant than uraninite and $FeU^{(V)}O_4$ NPs. Across five cells, HRTEM/SAED analyses identified $n = 231$ NPs in total, distributed as 128 $FeU^{(V)}O_4$ (55.4%), 93 uraninite (40.3%) and 10 pyrite NPs (4.3%) (Supplementary Fig. 8; Supplementary Data 1). In addition, TEM-based size measurements of $n = 160$ U nanoparticles allowed calculation of the averaged equivalent circular diameter (ECD), confirming that most NPs ranged between 2 and 3 nm, with a minority of larger particles extending the distribution tail (Supplementary Fig. 9; Supplementary Data 2).

The integration of EXAFS, HRTEM and HERFD-XANES allows for a detailed characterisation of U(V) species within the system. The results reveal the coexistence of at least two U(V) structural forms: U(V)-

**Table 1 | Shell fit EXAFS structural parameter for the ITFA isolated U species, the reference compounds $Na_4UO_2(CO_3)_3$, $UO_2(CO_3)_3^{4-}$, $UO_2(CO_3)_3^{5-}$ and for colloidal uraninite together with average radial distances (R) from literature**

| Path | CN | R/Å | DW/Å² | dE0/eV |
|---|---|---|---|---|
| U(V/VI) species, $S_0^2 = 0.96$ | | | | |
| $O_{ax}$ | 2.f | 1.85 (1) | 0.013 (1) | 7 (1) |
| MS U-$O_{ax(1)}$ -$O_{ax(2)}$ | 2./ | 3.69/ | 0.025/ | 7/ |
| $O_{eq}$ | 5.f | 2.456 (9) | 0.0084 (7) | 7/ |
| C | 2.5/ | 2.93 (4) | 0.014 (6) | 7/ |
| $Na_4UO_2(CO_3)_3$, $S_0^2 = 0.96$ | | | | |
| $O_{ax}$ | 2.0 (1) | 1.824 (2), 1.83 | 0.0019 (4) | 4.0 (4) |
| MS U-$O_{ax(1)}$ -$O_{ax(2)}$ | 2.0/ | 3.648/ | 0.0038/ | 4.0/ |
| $O_{eq}$ | 6.1 (3) | 2.408 (3), 2.40[97] | 0.0063 (5) | 4.0/ |
| C | 3.0/ | 2.884 (8), 2.90[97] | 0.004 f | 4.0/ |
| $UO_2(CO_3)_3^{4-}$ #, $S_0^2 = 0.96$ | | | | |
| $O_{ax}$ | 2.1 (1) | 1.799 (2), 1.80[45] | 0.0023 (4) | 3.5 (4) |
| MS U-$O_{ax(1)}$ -$O_{ax(2)}$ | 2.1/ | 3.598/ | 0.0046/ | 3.5/ |
| $O_{eq}$ | 5.9 (3) | 2.444 (4), 2.44[45] | 0.0069 (6) | 3.5/ |
| C | 3.0/ | 2.904 (8), 2.90[45] | 0.004 f | 3.5/ |
| $UO_2(CO_3)_3^{5-}$, $S_0^2 = 0.96$ | | | | |
| $O_{ax}$ | 2.1 (2) | 1.860 (4), 1.91[43] | 0.0070 (8) | 4.4 (6) |
| MS U-$O_{ax(1)}$ -$O_{ax(2)}$ | 2.1/ | 3.720/ | 0.0140/ | 4.4/ |
| $O_{eq}$ | 4.4 (3) | 2.481 (5), 2.50[43] | 0.0066 (7) | 4.4/ |
| C | 2.2/ | 2.94 (1), 2.93[43] | 0.004 f | 4.4/ |
| Uraninite-like species, $S_0^2 = 0.90$ | | | | |
| O | 9.2 (7) | 2.326 (6), 2.35, 2.36[46,47] | 0.016 (1) | −9.9 (7) |
| U | 4.3 (6) | 3.838 (5), 3.84, 3.86[46,47] | 0.0065 (8) | −9.9/ |
| Crystalline uraninite §, $S_0^2 = 0.90$ | | | | |
| O | 7.2 (4) | 2.316 (4), 2.34[34] | 0.0122 (8) | −8.5 (5) |
| U | 4.9 (5) | 3.840 (3), 3.85[34] | 0.0052 (5) | −8.5/ |

CN = coordination number, R = radial distance, DW = Debye-Waller factor, dE0 = shift in energy threshold, $S_0^2$ = amplitude reduction factor, f = fixed parameter, / = linked parameter. MS = multiple scattering path along the uranyl chain. Estimated standard deviations of the variable parameter in parenthesis. Spectrum of $UO_2(CO_3)_3^{4-}$ (#) and colloidal uraninite (§) taken from Rossberg et al.[45] and Veeramani et al.[34] but fitted in the shorter k-range of 3.00–11.95 Å⁻¹.

carbonate complexes such as $UO_2(CO_3)_3^{5-}$ and $FeU^{(V)}O_4$. EXAFS on wet pastes identified U(V)-carbonate species, whereas HRTEM confirmed $FeU^{(V)}O_4$ as a stable U(V)-bearing structure under reducing conditions. Both observations are consistent with the U(V) fractions quantified by HERFD-XANES and ITFA. In agreement with Crean et al.[31], our results indicate that the U(V) phase, attributed to $FeU^{(V)}O_4$, not only persists but may even become more stable over time under oxic conditions. The biogenic formation and high stability of $FeU^{(V)}O_4$ could therefore offer a major advantage over traditional U(IV)-based remediation strategies. This phase may represent a permanently stable form resistant to both reductive and oxidative remobilisation. Such an approach could potentially replace final treatment steps by enabling in situ immobilisation of U. Nevertheless, further studies are required to confirm its environmental compatibility.

The apparent discrepancy between EXAFS and HRTEM in detecting U(V) species reflects the complementary nature of the two techniques and their sensitivity to sample preparation. In EXAFS,

analysis of wet pastes preserved part of the aqueous phase, potentially stabilising U(V)-carbonate complexes and increasing their proportion. In contrast, ethanol dehydration prior to HRTEM, required for resin embedding of biological samples, reduces the aqueous content and may destabilise these complexes. This could explain their lower visibility in HRTEM and the predominance of $FeU^{(V)}O_4$ and $UO_2$ nanoparticles. Nevertheless, the complementary results provide a coherent speciation model, particularly as the proportion of $FeU^{(V)}O_4$ continues to increase with prolonged incubation, indicating an ongoing process.

Although both species share the same oxidation state, they exhibit key structural differences. U(V)-carbonate corresponds to a complex in which the pentavalent uranyl ion is coordinated by three carbonate ions. In contrast, $FeU^{(V)}O_4$ is a complex in which U(V) interacts with iron to form a more stable structure. Nevertheless, the identification of $FeU^{(V)}O_4$ through HRTEM confirms that this structure dominates the system and accounts for a substantial proportion of the NPs formed. These findings demonstrate that U(V) can exist in multiple structural forms depending on local conditions, such as the presence of Fe, illustrating the complexity of U speciation in the studied system. This underscores the need to develop improved analytical standards for specific U species and to use complementary techniques to better understand the dynamics of U(V) and its role in biogeochemical processes.

Advanced spectroscopy and high-resolution microscopy provide direct evidence for the formation of U(V) species under environmentally relevant mine-water conditions. These results demonstrate both biotic and abiotic formation of U(V) species at very low U concentrations (1 mg mL⁻¹), highlighting their relevance for bioremediation. However, the identification of U(V) in distinct structural forms, such as U(V)-carbonate and $FeU^{(V)}O_4$, suggests that specific bio-geochemical conditions can stabilise U(V). The dominance of U(IV) as uraninite in the final stages reflects the thermodynamic stability of this phase under the given conditions. These findings underscore the importance of coupling microbial reduction pathways with geochemical interactions to understand U cycling in natural systems.

**Integrated biogeochemical and U speciation characterisation**

The 16S rRNA gene sequencing results showed that glycerol biostimulation induced distinct changes in the microbial community in the microcosms (B4 and B8), (Supplementary Fig. 10 and Supplementary Data 3), compared to the initial microbial community in the Schlema-Alberoda mine water[3]. In the initial microbial community, dominant phyla such as Proteobacteria, Campylobacterota, Patescibacteria, Verrucomicrobiota and Nitrospirota, including nitrate reducers, sulphur oxidisers (genus *Sulfuricurvum, Sulfurovum, Sulfurimonas*), potential sulphate reducers (class Thermodesulfovibrionia and family Desulfobulbaceae), iron oxidisers (*Gallionella*) and potential iron reducers (family Rhodocyclaceae) was identified in the Schlema-Alberoda mine water[3].

The glycerol-biostimulated enriched bacterial community from the microcosms B4 and B8 (at day 130) comprised several taxa associated with anaerobic glycerol fermentation, such as an unidentified genus from the Propionibacteriaceae family (16.13%) and *Propionivibrio* (2.82%)[20,54]. Other bacteria such as *Berkelbacteria* (15.33%), an unidentified genus from the Christensenellaceae family (13.06%), an unidentified genus from the Prolixibacteraceae family (6.65%), Absconditabacteriales SR1 (3.86%) and *Brevinema* (2.85%) may be involved in both reductive and oxidative pathways of glycerol metabolism. These pathways produce metabolic intermediates such as propionate, acetate, lactate, butyrate, ethanol, $H_2$ and $CO_2$[55]. Although these fermentative taxa are not directly responsible for U reduction, their glycerol metabolism generate short-chain fatty acids (e.g. acetate, lactate), hydrogen gas ($H_2$), which can serve as electron donors for U(VI)-reducing bacteria, including sulphate-reducing (e.g. *Desulfovibrio*) and metal-reducing (e.g. *Geobacter*) species[19].

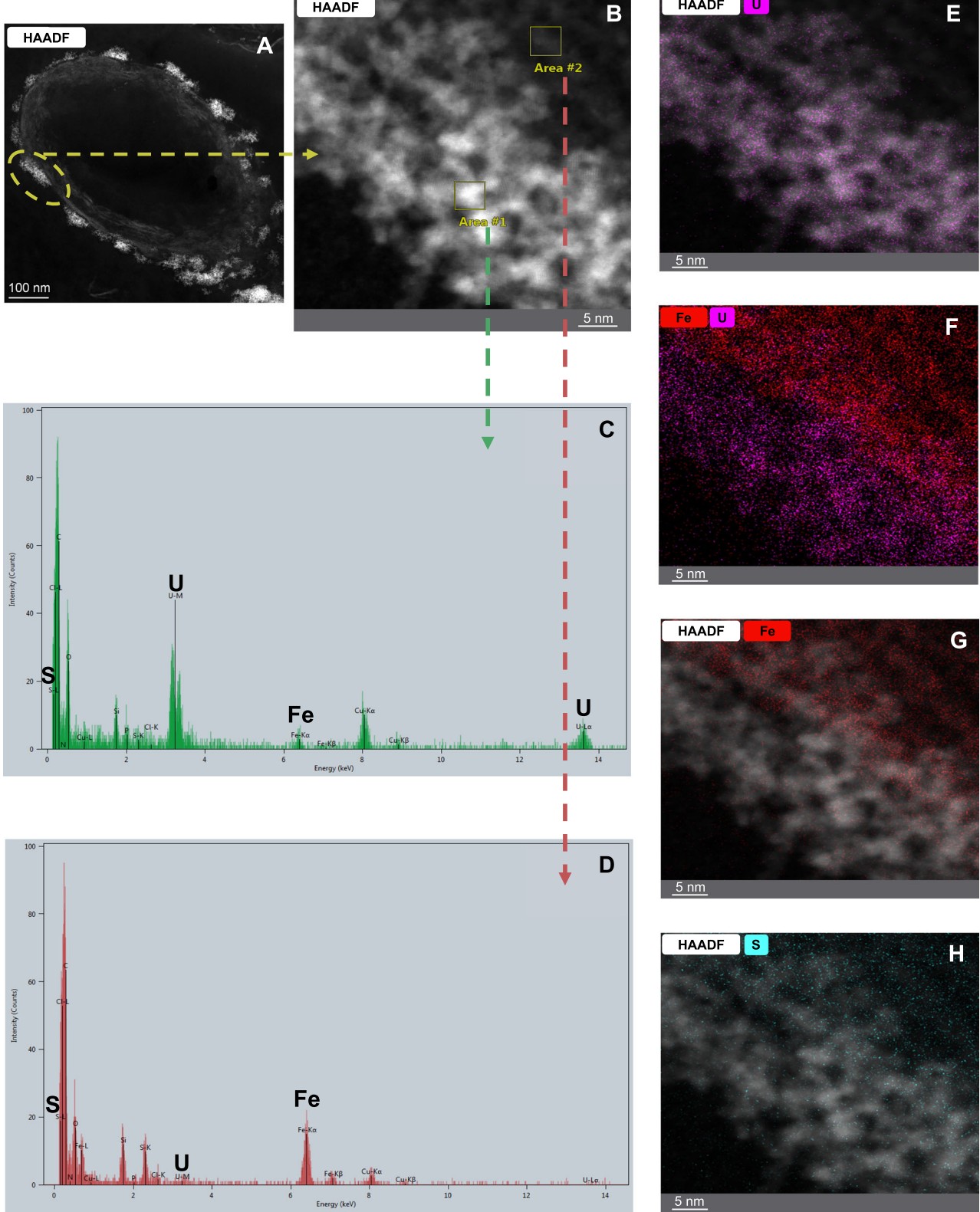

**Fig. 3 | HAADF-STEM imaging and elemental mapping of U precipitates.** HAADF-STEM micrographs of a thin section of the collected black precipitate collected at the sampling point corresponding to an approximately 90% decrease in dissolved U(VI) concentration in the aqueous supernatant (reached after ~55 days of incubation) show electron-dense agglomerates produced during incubation and induced by biostimulation of the native microbial community with glycerol (**A**, **B**). EDXS spectra (**C**, **D**) and elemental distribution maps (**E–H**) display the elemental composition of U, Fe and S.

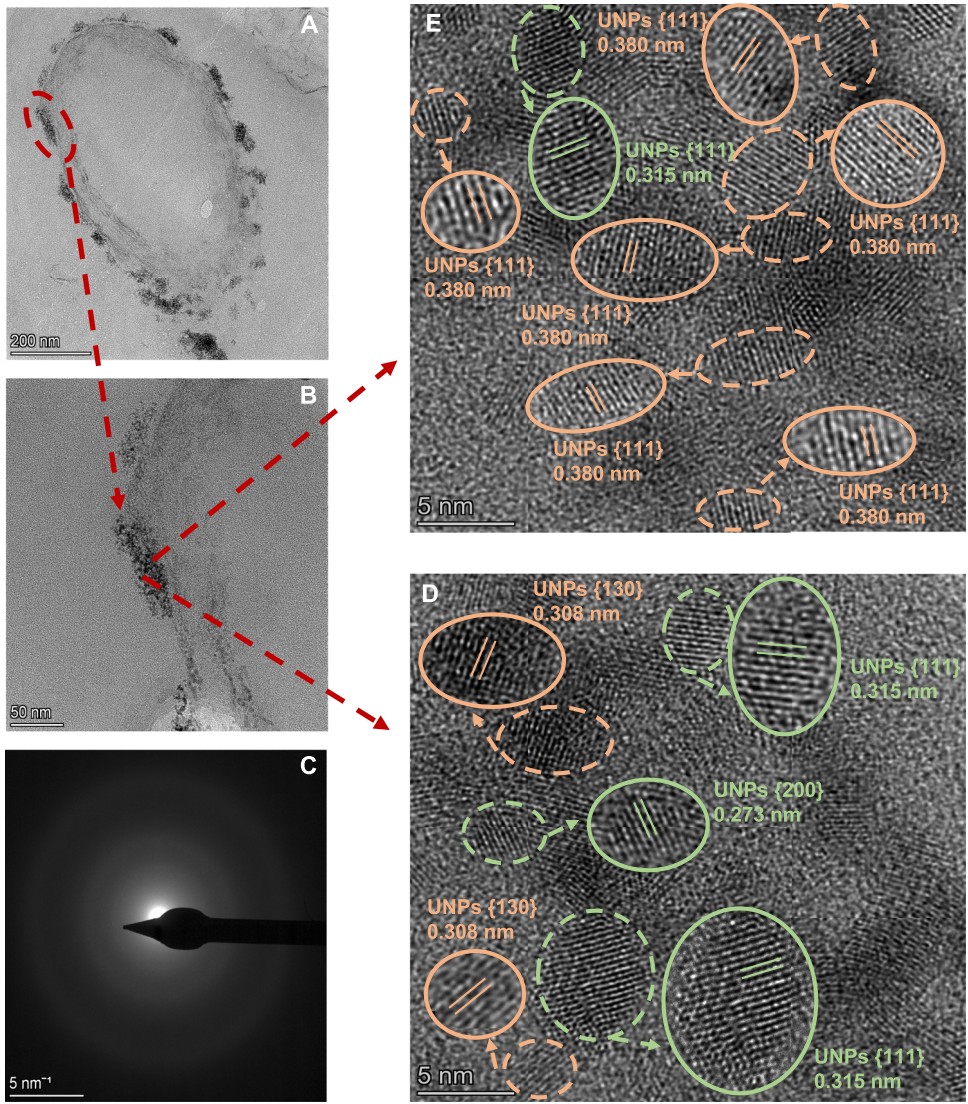

**Fig. 4 | HAADF-STEM and HRTEM imaging of U nanoparticles.** HAADF-STEM micrograph of U nanoparticles (UNPs) from the collected black precipitate shows electron-dense clusters formed during incubation and induced by biostimulation of the native microbial community with glycerol (**A**, **B**). Enlarged SAED pattern (**C**) and HRTEM images (**D**, **E**) correspond to the interior of the accumulation and reveal several aggregated UNPs. Lines drawn in the magnified circles indicate lattice spacings corresponding to crystallographic planes. Uraninite NPs are marked in green and FeU$^{(V)}$O$_4$ NPs in orange circles.

The SRB community in the analysed microcosms was dominated by *Desulfobulbus* (14.14%) and *Desulfovibrio* (1.05%), both well-known sulphate-reducing genera capable of oxidising a range of organic substrates, including glycerol, coupled to sulphate reduction[56–58]. The signature of these bacterial taxa was minor in the initial microbial community in the Schlema-Alberoda mine water, suggesting that their enrichment resulted from the microcosm conditions applied in this experiment[3]. Although we did not conduct the metatranscriptomic profiling of the microcosm microbial communities, the differential expressed genes from the original Schlema-Alberoda mine water (time 0) revealed complete gene sets for both dissimilatory sulphate reduction (DSR) (*sat, aprAB, dsrAB*) and assimilatory sulphate reduction (ASR) (*sat, cysDNC, cysC, cysH, cysJI, sir, PAPSS*) (Supplementary Table 3, Supplementary Fig. 11). The end metabolite of both pathways is H$_2$S based on KEGG (sulphur metabolism, map 00920). These genes are conserved markers of sulphur metabolism in *Desulfovibrio* and other SRB, where *sat* gene act as a central link between the assimilatory and dissimilatory pathways[59].

The identification of gene transcripts related to complete DSR and ASR suggested that the sulphate reducers in the original mine water were metabolically active in situ prior to our microcosm setup. Together with the observed decrease in sulphate content in the microcosms, our data indicate that *Desulfobulbus* and *Desulfovibrio* reduced sulphate to H$_2$S, which could potentially contribute to indirect abiotic U(VI)[36,37]. On the other hand, direct enzymatic U(VI) reduction has been reported in certain species of *Desulfovibrio*[35,60]. As a result, U(VI) reduction in the microcosms likely involved a combination of direct enzymatic pathways and indirect abiotic contributions mediated by biogenic H$_2$S. In parallel, both *Desulfovibrio* and *Desulfobulbus* can reduce Fe(III) to Fe(II), which could subsequently abiotically reduce U(VI) through electron transfer[35,37].

The presence of U(V) as an intermediate in microbial U(VI) reduction has been previously documented under controlled laboratory conditions, mainly using pure cultures of IRB and SRB[15–17,61]. Recent studies highlighted the role of multiheme c-type cytochromes in the reduction of U(V)-dpaea and U(VI)-dpaea complexes by *Shewanella oneidensis* MR-1 and emphasised their importance in U bioremediation[62]. Together, these studies suggest that U(V) can be generated via a one-electron transfer mechanism, followed by disproportionation to U(VI) and/or U(IV) due to its instability.

In the glycerol-amended microcosms, glycerol appeared to favour the growth of the above-mentioned fermentative microorganisms, which converted it into low-molecular-weight metabolites. These metabolites subsequently stimulated the activity of sulphate- and metal-reducing bacteria. The activity of these groups was reflected in concomitant geochemical changes in the microcosms, including progressive sulphate and iron removal and a marked decrease in redox potential (Supplementary Figs. 2 and 3). Collectively, these processes stablished strongly reducing conditions and generated additional abiotic reductants (e.g. Fe (II)), which, together with microbial enzymatic pathways, facilitated U(VI) reduction.

Under these conditions, spectroscopic analyses revealed that U(VI) reduction proceeded not directly to U(IV), but via U(V) as an intermediate. Although disproportionation of U(V) cannot be excluded, our U $M_4$-edge HERFD-XANES data reveal substantial fractions of U(V) persist within the precipitates. Remarkably, this stability persisted for 130 days, even under oxic exposure (Fig. 1A–D). The integration of microbial, geochemical, microscopic and spectroscopic results thus shows that U(VI) reduction in mine waters does not lead exclusively to U(IV), but also leads to the stabilisation of U(V) under environmentally relevant conditions. This expands the classical U(VI)/U(IV) paradigm and highlights an additional pathway of U immobilisation with direct relevance for bioremediation scenarios.

## Long-term environmental implications of U(IV)/U(V) formation

Bioreduction offers a potentially effective alternative or complementary remediation strategy[2,12]. However, the re-oxidation of U-reduced products must be considered, as it can compromise their long-term stability[11,18]. Biogenic uraninite can be easily re-oxidised due to pH shifts, $E_h$ fluctuations, increased nitrate levels, or oxygen exposure[8,36]. Uraninite NPs are particularly vulnerable because of their high surface reactivity and mobility in groundwater[63]. Consequently, assessing their stability in the presence of oxidants like $O_2$ is crucial for optimising U bioreduction strategies.

Building on these findings, we evaluated the environmental implications of U(V) persistence and U(IV) re-oxidation under mine-water conditions. HERFD-XANES and ITFA analyses revealed the coexistence of U(IV) and U(V) species within the bioreduced solids, reflecting a dynamic but stabilised redox system. Over 130 days of incubation, U(V) remained detectable under anoxic conditions and persisted even after 4 weeks of oxic exposure, indicating a remarkable resistance to re-oxidation. The results (Fig. 1D, E) quantified partial transformation of U(IV) (7%) to U(VI) (40%) while U(V) accounted for 53% of total U after oxic incubation, underlining both the susceptibility of biogenic uraninite to oxidation and the stability of U(V). This persistence suggests that complex microbial–mineral interactions can stabilise intermediate U oxidation states under realistic environmental conditions.

Our findings demonstrate that U(V) persistence provides an additional mechanism for limiting U mobility beyond the conventional U(VI)/U(IV) framework. Acting as a buffer state, stabilised U(V) can delay U(VI) remobilisation even if U(IV) is partly re-oxidised, with important implications for the long-term stability of bioremediation products.

This is particularly relevant because, until now, the persistence of U(V) under environmentally relevant conditions has rarely been demonstrated and has seldom been a central focus of investigation. Rather than relying on a complete and potentially reversible conversion to U(IV), remediation may therefore benefit from the coexistence of U(IV) and U(V), both contributing to U immobilisation.

In this context, it is important to consider that U immobilisation can also occur through mineralogical pathways in addition to microbiological processes. Fe-rich clays, for instance, have been shown to retain U(VI) through interlayer confinement, where uranyl ions intercalate or become sequestered within expandable clay interlayers, leading to persistent retention in nanoconfined microenvironments[64–66].

Under reducing conditions, structural Fe(II) present in clay interlayers may act as an abiotic reductant, promoting the reduction of U(VI) predominantly to U(IV), which may subsequently become strongly retained within interlayer domains or associated mineral sites[64,67,68]. In addition, recent observations suggest that U(V) may form transiently on internal or external clay surfaces during Fe(II)-mediated reduction processes[69], although structurally confined U(V) has not yet been demonstrated.

Our study does not address clay minerals or interlayer processes. Instead, it reveals a distinct immobilisation pathway in which reduced U species (U(IV) and U(V)) are stabilised at microbe–mineral interfaces and within biogenic Fe–U phases under environmentally relevant mine-water conditions. While mineral interlayer retention and surface- or nanoparticle-associated stabilisation can coexist in heterogeneous natural systems, they are governed by fundamentally different redox drivers, mineral structures and physicochemical constraints. Recognising these distinctions is essential for evaluating U(IV) and U(V) stability and their long-term immobilisation in complex subsurface environments for safety analysis.

Although the precise mechanism remains unresolved, the formation of U(V) is likely initiated by enzymatic one-electron transfer reactions characteristic of microbial U(VI) reduction, subsequently influenced by abiotic and structural stabilisation processes. The persistence of U(V) under environmentally relevant conditions appears to result from multiple, interconnected stabilisation pathways. Complexation with biogenic ligands derived from microbial activity may reduce U(V) reactivity in solution. Incorporation into Fe-containing nanoparticles such as $FeU^{(V)}O_4$ can provide structural stabilisation within solid phases. In addition, adsorption onto microbial biomass or mineral surfaces may create protective microenvironments that limit disproportionation and oxidation. These combined processes are consistent with previous reports of U(V) stabilisation in laboratory and other systems[15,17,31]. However, in the present study, we were able to confirm this under realistic environmental conditions in mine water. Overall, these findings provide the first direct experimental evidence for persistent U(V) in mine-water microbial systems and underline its relevance for the environmental fate of U and future bioremediation approaches.

## Implications for U immobilisation and remediation

This study demonstrates the effectiveness of glycerol as an electron donor to stimulate microbial U(VI) reduction in mine-water microcosms containing low U concentrations (1 mg mL$^{-1}$) under neutral-alkaline and carbonate-rich conditions. Spectroscopic and microscopic analyses confirm the simultaneous formation of U(IV), mainly associated with biogenic uraninite nanoparticles, and stabilised U(V) in two distinct forms, a U(V)-carbonate complex and $FeU^{(V)}O_4$, located on bacterial surfaces. The persistence of U(V) under anoxic conditions for 130 days and its detection after 4 weeks of oxic exposure provide direct evidence of biogenic U(V) species under environmentally realistic mine-water conditions, underscoring their role as intermediates contributing to U immobilisation.

Microbial community analysis revealed enrichment of fermenters (e.g. *Propionibacteriaceae*, *Propionivibrio*) and SRB (*Desulfobulbus*, *Desulfovibrio*), supporting complementary metabolic pathways. Fermenters generate electron donors from glycerol degradation that promote SRB activity. However, many SRB are also known to use glycerol directly. These bacteria contribute directly to U(VI) reduction and help maintain reducing conditions, while indirect abiotic contributions involving biogenic sulphide or Fe(II) remain plausible but cannot be assessed with our dataset. Together, these processes sustain U(VI) reduction and stabilisation. The convergence of microbiological, geochemical and spectroscopic evidence indicates that U(V)

formation and stabilisation arise from interactive processes rather than a single mechanism.

From a practical perspective, remediating low U contaminated waters under neutral-alkaline conditions is challenging because strong carbonate complexation maintains U(VI) in solution and increases the redox demand, while reduced species remain vulnerable to re-oxidation. In this context, the persistence of U(V) as $FeU^{(V)}O_4$ offers a stable immobilisation pathway capable of withstanding redox fluctuations and supporting long-term U immobilisation.

These findings underscore the relevance of U(V) as a persistent intermediate in bioremediation settings. Although derived from a single geochemical scenario, the processes identified here are broadly applicable to other contaminated waters, where the distribution of U(V) and U(IV) will be shaped by pH, the availability of competing electron acceptors, Fe mineralogy and organic ligands. Future studies should assess how these factors interact to govern U(V) persistence across environmentally relevant conditions. Within this framework, our study provides new insights for understanding U(V) persistence and for developing more robust remediation strategies.

## Methods

### Elaboration and monitoring the chemistry of glycerol-amended mine water microcosms

Water samples for the microcosm experiments were obtained from the inlet of the treatment plant at the Wismut GmbH Schlema-Alberoda mine[3]. Anoxic microcosms were established to evaluate the bioremediation potential of the native U-reducing microbial community. Two-litre serum bottles were filled with fresh mine water and supplemented with 10 mM glycerol, the optimal electron donor for U reducers[3]. Control microcosms included mine water (SAC) without glycerol and sterilised mine water with glycerol (ASA + G). Each condition was prepared in triplicate. All microcosms were incubated in the dark at $28 \pm 2\,°C$ for 130 days.

Physicochemical parameters of the mine water were monitored weekly using inductively coupled plasma mass spectrometry (ICP-MS) and high-performance ion chromatography (HPIC). To assess predominance fields of possible U species, a Pourbaix diagram was generated using Geochemist's Workbench (v18.0.3), based on the PSI Chemical Thermodynamic Database 2020[38,39].

Further details on microcosm setup, analytical procedures, thermodynamic parameters and calculation methods are provided in the Supplementary Methods 2.

### X-ray absorption spectroscopy characterisation of U products

Sampling was performed at defined incubation times, selected to correspond to conditions at which the dissolved U(VI) concentration in the aqueous supernatant had decreased by approximately 20%, 60% and 90%, as determined from ICP-MS measurements. Based on the time-course data of dissolved U(VI) concentrations and redox potential (Supplementary Fig. 2), these conditions were reached after ~10 days (day 10), ~30 days (days 20–40) and ~55 days (days 52–60) of incubation, respectively. This sampling strategy ensured that the spectroscopic analyses targeted distinct chemical and redox states of the system at the time of sampling. The samples were centrifuged ($4020 \times g$, 15 min, Hettich EBA 21, Germany) and the black precipitates were prepared under strictly anoxic conditions for U $M_4$-edge HERFD-XANES and U $L_3$-edge EXAFS. Immediate cryogenic freezing preserved U speciation for synchrotron radiation-based measurements. To assess stability, a portion of the pellet obtained from the microcosms at the sampling point corresponding to ~90% decrease in dissolved U(VI) (after ~55 days of incubation) was placed on the sample holder. It was then exposed to ambient air (~21% $O_2$) in a closed laboratory at $21 \pm 1\,°C$ for 4 weeks, without active aeration. The holder remained open to allow unrestricted gas exchange but was covered to prevent particulate contamination. After exposure, the sample was sealed and subjected to HERFD-XANES analysis.

Measurements were performed at BM20 Rossendorf beamline (ROBL) of the European Synchrotron Radiation Facility (ESRF) in Grenoble (France)[70] where the storage ring was operated in the multi-bunch filling mode at 6 GeV with a 200 mA current. HERFD-XANES samples were placed as wet pastes in 1-mm deep holders, single-confined with 13-µm Kapton foil. EXAFS samples were transferred into 3-mm polyethylene holders, double-confined with 13-µm Kapton tape and polyethylene. All samples were frozen in liquid $N_2$ and measured under cryo-conditions.

HERFD-XANES measurements were performed at the U $M_4$-edge[71] (3728 eV), and fluorescence EXAFS measurements at the U $L_3$-edge[72] (17168 eV). HERFD spectra were recorded using a Johann-type X-ray emission spectrometer in a vertical Rowland geometry[73–75] equipped with a silicon drift X-ray detector (©Ketek). The incident energy was selected using a Si(111) double-crystal monochromator. Two Si mirrors before and after the monochromator were used to collimate the beam and reject higher harmonics. The incident energy was calibrated using HERFD spectra of reference compound, i.e. the maximum energy position of U HERFD-XANES was set at 3725.0 eV (U $M_4$-edge, $UO_2$ reference). The beam size was estimated to be ~30 µm (vertically) by ~2 mm (horizontally). The X-ray emission spectrometer was equipped with five Si(220) crystal analysers with a 1 m bending radius. The spectrometer was tuned to the maximum of the U $M_β$ emission line (3339.8 eV). The corresponding Bragg angle was 75. A helium gas-filled bag was placed to fill the optical path sample-crystal analysers-detector to reduce the absorption of the fluorescence signal by air. The energy resolution was estimated to be ~0.7 eV. The HERFD-XANES U $M_4$-edge spectra were recorded with 0.2 eV step and the counting time of 3 s per point. Each individual spectrum was ~6 min of duration. To increase signal-to-noise ratio, 4–10 spectra per sample were collected and averaged. The U(IV) and U(VI) reference spectra were obtained from $UO_2$ and uranyl(VI)-nitrate measured during the same beamtime and used to calibrate the corresponding energy positions. The U(V) assignment (~3726.5 eV) was validated against the $UMoO_5$ reference spectrum[22].

In the case of U $L_3$-edge EXAFS measurements the white X-ray beam was monochromatized by using a Si(111) double crystal monochromator in channel cut mode, while two Rh-coated mirrors reduced the higher harmonics. For each sample the fluorescence signal of the $Lα_{1,2}$ line was accumulated by using a 18-element Ge-detector and the K-edge absorption spectrum of a Y metal foil was measured simultaneously for energy calibration. The incident photon flux and the absorption of the Y metal foil was measured by using gas-filled ionisation chambers. Per sample 6–9 energy scans were accumulated and averaged in order to receive a sufficiently high signal-to-noise ratio as needed for further data analysis. The samples were measured under cryogenic conditions by using a closed cycle He-cryostat. The corresponding raw U $L_3$-edge XAS spectra are shown in Supplementary Fig. 12. For the calculation of the photoelectron wave vector ($k$) the ionisation potential at the U $L_3$-edge was arbitrarily set to 17185 eV. We used the EXAFSPAK software for data processing, including energy calibration, averaging of multiple sample scans, X-ray absorption background correction, EXAFS signal isolation and structural model fitting[76]. The ab-initio scattering code FEFF8.20[77] was used for calculating theoretical amplitude and phase scattering functions, while liebigite[78] and uraninite[79] served as structural models. Assuming the presence of coexisting U(IV), U(V) and U(VI) species, we applied ITFA[80] for the mathematical decomposition of the spectral mixtures into pure U component spectra and their fractions in the data. The corresponding comparison between the experimental U $M_4$-edge HERFD-XANES spectra and the ITFA-derived model fits is shown in Supplementary Fig. 13. Detailed information on U oxidation state quantification using ITFA can be found in the Supplementary Methods 1.

## High-resolution transmission electron microscopy analysis

Crystallographic and cellular localisation of U reduction products were analysed using HAADF-STEM (FEI TITAN G2 80-300) at the Centro de Instrumentación Científica (CIC, University of Granada, Spain), equipped with EDXS and SAED. The black precipitate, collected at the sampling point corresponding to an approximately 90% decrease in dissolved U(VI) concentration in the aqueous supernatant (after ~55 days of incubation), was fixed in 2.5% glutaraldehyde (10% PBS), embedded in Epon 812 resin, and sectioned (75–80 nm) using a diamond knife on an ultramicrotome (Reichert Ultracut S, Germany). Sections were placed on copper grids, carbon-coated and examined at 200 kV with liquid nitrogen anti-contamination. EDXS visualised U, Fe and S distribution in bacterial membranes.

Nanoparticles observed by HRTEM were quantified by two complementary analyses, comprising particle size measurements ($n = 160$) and crystallographic plane assignment from d-spacing values ($n = 231$). Further methodological details, including the calculation of ECD[81] and histogram construction are provided in Supplementary Methods 3. SAED measured the d-spacing of 231 NPs in the membrane, assigning them to crystallographic planes via the American Mineralogist database (http://rruff.geo.arizona.edu).

## 16S rRNA-based microbial community analysis for microcosms

Microbial community composition was analysed at the end of the 130-day incubation, when the concentration of U(VI) in the supernatant had decreased by more than 90% in the glycerol-amended microcosms. The water samples from two representative microcosms (B4 and B8), which exhibited similar geochemical conditions and U(VI) removal efficiencies, were selected for DNA extraction and 16S rRNA gene amplicon sequencing. For each microcosm, 400 mL of water were sampled independently three times ($n = 3$), and the biomass from each sample was collected on sterile 0.45- and 0.20-µm membrane filters (MF-Millipore®, Germany) and stored at −20 °C. Each filter was quartered, and sections were placed in 5 mL sterile screw-cap tubes (DNeasy Soil Kit, QIAGEN, Germany) for DNA extraction following the manufacturer's instructions. DNA integrity was verified via 0.75% agarose gel electrophoresis, and concentration measured with a Qubit Fluorometer 4.0 (Thermo Fisher Scientific, USA). DNA from filter sections was pooled into a 1.5-mL low-retention tube, forming a single replicate. Samples were stored at −20 °C until amplification. The bacterial 16S rRNA gene was amplified targeting the V3-V4 regions[3]. PCR, library preparation, and sequencing (Illumina MiSeq) were conducted at STAB-VIDA (Caparica, Portugal; https://www.stabvida.com/es).

For bioinformatic analysis, FastQC was used for quality control of raw sequences[82]. Reads were processed with QIIME 2 v2024.10[83]. DADA2 was applied for read denoising, including trimming, truncation, dereplication and chimera removal[84], yielding amplicon sequence variants. Taxonomy was assigned using a scikit-learn Naïve Bayes classifier trained on SILVA 138 (QIIME) with reference sequences clustered at 99%[85].

## Metatranscriptomic analysis of Schlema-Alberoda mine water

To assess the functional genes of the native microbial community prior to biostimulation, metatranscriptomic profiling was performed on mine water collected from the Schlema-Alberoda site. The U concentration of mine water of Schlema-Alberoda was 1 mg mL$^{-1}$. To identify active metabolic processes within the mine-water community via differential gene expression (DGE), metatranscriptome profiling was also conducted on Pöhla mine water. This site is a former U-mining location in Saxony where the U concentration was 0.01 mg mL$^{-1}$, approximately 100 times lower than in Schlema-Alberoda[3,5]. To extract RNA from both mine water, 800 mL of fresh mine water was filtered sequentially through a sterile 0.45- and 0.20-µm membrane filters to collect the microbial biomass (MF-Millipore®, Germany). The membrane filter was promptly frozen in liquid nitrogen and stored at −80 °C. The total RNA from each mine water was extracted individually ($n = 3$) using the RNeasy PowerWater Kit (QIAGEN, Germany) including an on-column DNase digestion step. RNA concentration was determined with a Qubit Fluorometer 4.0 (Thermo Fisher Scientific, USA) using the RNA HS Assay Kit, and RNA integrity was assessed with an Agilent 2100 Bioanalyzer (Agilent Technologies, USA).

RNA from each replicate was subjected to rRNA depletion, library preparation and then sequencing on an Illumina NovaSeq platform (150 bp paired-end reads) (STAB-VIDA, Caparica, Portugal; https://www.stabvida.com/es).

Downstream bioinformatic analysis was conducted as previously described by Wei et al.[86]. In brief, sequencing adaptors and low-quality bases (QS < 30) were trimmed using Cutadapt v4.2[87] and Trimmomatic v0.39[88], respectively, prior to remaining rRNA reads removal with SortMeRNA v4.3.4. with Silva 138 SSURef and LSRRef[89]. Clean reads from all samples were assembled de novo with Trinity v2.14.0[90] to generate a single assembly. The read counts were estimated using Bowtie2 v2.5.1[91] and then quantified using featureCounts (Subread v2.0.6)[92]. The edgeR software package was used for analysing DGE between Schlema-Alberoda and Pöhla communities[93]. The open reading frames (ORFs) of the assembled transcripts were predicted using Prodigal v2.6.3[94]. Functional annotation, particularly metabolic pathways, was achieved applying protein sequences of these genes against the KEGG Orthology (KO) database using GhostKOALA[95,96].

## Statistics and reproducibility

Microcosm experiments were conducted in biological triplicates unless otherwise stated. No data were excluded from the analyses. The experiments were not randomised and the investigators were not blinded to allocation during experiments and outcome assessment. Where applicable, data are reported as mean ± standard deviation. Spectroscopic measurements were repeated and averaged to improve signal-to-noise ratios.

## Data availability

The raw sequencing data generated in this study have been deposited in the NCBI Sequence Read Archive (SRA) under accession numbers PRJNA1243298 (16S rRNA gene sequences) http://www.ncbi.nlm.nih.gov/bioproject/1243298 and PRJNA1358374 (metatranscriptomic data) http://www.ncbi.nlm.nih.gov/bioproject/1358374. Additional experimental, geochemical and spectroscopic datasets supporting this study are available in the RODARE repository under https://doi.org/10.14278/rodare.3546. Source data underlying the figures are provided with this paper, and additional data supporting the findings of this study are available in the Supplementary Information. The data supporting the findings of the study are included in the main text and supplementary information files. Raw data can be obtained from the corresponding author upon request.

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

## Acknowledgements

The authors gratefully acknowledge WISMUT GmbH (Germany) for providing samples for the experiments, sharing site-specific remediation information, and offering technical support during the sampling campaigns. We also thank the staff at the Helmholtz-Zentrum Dresden-Rossendorf, especially Sindy Kluge for coordinating microbiological and general sample handling, Sabrina Beutner for performing ICP-MS analyses, Sylvia Schöne for HPIC measurements, and Stephan Weiss, Rahel Bertheau, and Susana Jiménez for their technical assistance. Additionally, we appreciate the support from María del Mar Abad Ortega for her assistance in the measurement and interpretation of microscopy samples and Daniel García Muñoz Bautista for the preparation of samples for microscopy, both from the Centro de Instrumentación Científica at the Universidad de Granada. The research presented in this article has received funding from the Plan Propio de Investigación programme of the University of Granada, managed by the Oficina de Transferencia de Resultados de Investigación (OTRI), and the RadoNorm project of the Euratom Research and Training Programme 2019–2020 under grant agreement No. 900009 (M.L.M. and A.M.N.-P.). Additional support to A.M.N.-P. was provided through the mobility grants programme of the European Radioecology Alliance (ALLIANCE) and the University of Granada (Plan Propio de Investigación, P10 Programme for Research Stays).

## Author contributions

E.K.-B, M.L.M., J.R. and A.M.N.-P. conceived the study. A.M.N.-P. M.L.M. and E.K.-B. designed and carried out the microcosm experiments and chemical analyses. F.B. contributed to uranium speciation modelling based on thermodynamic calculations. A.M.N.-P. and M.L.M. conceived and performed DNA extraction, 16S rRNA gene sequencing and microscopy analyses. M.L.M., A.M.N.-P. and S.T.-S.W. conceived and performed RNA extraction, analysis and contextualisation. K.O.K., E.B., A.R., E.K.-B. and A.M.N.-P. conducted HERFD-XANES and EXAFS measurements at the ROBL beamline (ESRF). K.O.K., E.B., A.R., A.M.N.-P. and E.K.-B. contributed to sample preparation and beamtime support. K.O.K., E.B. and A.R., processed spectroscopic data and performed ITFA modelling. A.M.N.-P. wrote the original draft of the manuscript with input from all co-authors. E.K.-B., K.O.K., J.R. T.S. and M.L.M. supervised the project and provided critical revision of the manuscript. The acquisition of funding was led by M.L.M., A.M.N.-P. and T.S.; resources and technical infrastructure were provided by T.S., A.K., J.R. and M.L.M.

## Funding

## Competing interests

The authors declare no competing interests.
