## [Transparent Peer Review file · Nature Communications]

Pentavalent and Tetravalent Uranium Formation via Glycerol-Stimulated Bacteria in Mine Water

Corresponding Author: Dr Antonio Newman Portela

Version 1:

Reviewer comments:

Reviewer #1

(Remarks to the Author)

The manuscript presents a combination of advanced spectroscopic, microscopic, and geochemical techniques to demonstrate that biological U(VI) reduction proceeds through, and partially terminates in, a pentavalent intermediate (FeU(V)O₄) that survives for weeks under oxic, near-neutral conditions. Nevertheless, the significance is not well articulated and many aspects of microbiology, mechanistic interpretation, and data presentation require substantial strengthening. Particularly, the microbial analysis should be improved significantly. Regardless, I still think, this observation is quite novel and potentially impactful for uranium remediation science. I therefore recommend major revision. If the authors can significantly improve the manuscript, then I will reconsider.

Specific comments:

- 1.Line 26, Abstract: In the first sentence I do not see the significance or importance of the issue.
- 2.Line 30: Is 16S rRNA analysis sufficient? Please justify.
- 3.Line 36: It is better to add "To the best of our knowledge" before "first report."
- 4.Introduction: The structure is quite messy. Lines 52–65 devote too much space to the situation in Germany while offering only a few words on the global context. What about the international situation?
- 5.Lines 66–68: Which physicochemical methods were used, and why did they fail? Provide comparative evidence; simply stating that bioremediation is effective is overly assertive.
- 6.Line 72: Clarify whether biomineralisation includes enzymatic reduction. The mechanism of uranium bioreduction is not explained in sufficient depth.
- 7.Lines 75–79: The logic is difficult to follow. The definition of waste glycerol is vague. Why choose glycerol when other waste products can serve as carbon sources? Is it more efficient? Is there large-scale waste glycerol production? Have you calculated the cost, including collection and transportation, compared with commercial carbon sources? Is carbon supply the main cost driver in bioremediation? This is unclear.
- 8.Lines 80–89: The first paragraph already states that U(V) forms and is often overlooked during reduction from U(VI) to U(IV). Repeating this here is redundant; consider merging.
- 9.Lines 97–98: The mere absence of previous studies does not automatically establish importance; explain why this gap matters.
- 10.Lines 100–108: Again redundant; please streamline.
- 11.Lines 109–129: State the study's aim succinctly and concisely outline the research content rather than listing specific methods and instruments. Indicate the perspectives you address, how you meet your aim, and the overall significance (which is currently unclear).
- 12.Results and Discussion, Line 151: What is the dash for? Please clarify.
- 13.Figure 3: Explain how you calculated the mass balance of uranium. Do the three oxidation states sum exactly to 100 %?
- 14.Several paragraphs are overly long, making it easy for readers to lose interest; consider breaking them into shorter, coherent units.
- 15.The spectroscopic and microscopic results are very interesting, but the subtitle promises an integrated discussion that is not delivered. Strengthen the overall synthesis of results and discussion.
- 16.My major concern: You present strong physicochemical data, yet for the microbial aspect you rely only on 16S rDNA. This technique mainly provides relative abundance and offers limited functional insight. You need greater depth to explain why and how microbes form U(V). Consider adding metagenomics, transcriptomics, or quantitative PCR.
- 17.Lines 373–381: Fermentative bacteria may not necessarily be related to uranium remediation; please clarify their relevance.

18. Abrupt mention of SRB: Introducing sulphate-reducing bacteria here feels abrupt. Provide background for readers and mention their relevance earlier in the Introduction.

19. Lines 382–388: These are well-established facts; condense or cite authoritative reviews.

20. Lines 395–396: Coupled S, Fe, and U biogeochemical cycles are a vast topic; omit this unless you treat it in detail.

21. I do not see a thorough discussion linking microbial results with the earlier physicochemical findings; please integrate these aspects more deeply.

22. Lines 419–445: The manuscript proposes U(V) formation, but its stability is discussed mainly with reference to literature. More importantly, the study's aim is still unclear. While the formation of U(V) is intriguing, what is its broader significance? You note that U(V) is more stable than expected and may aid U(VI) bioremediation, but the mechanism needs explanation. Clarify this for readers.

Other concerns and comments

1. The key advance is the persistence of U(V) after four weeks of air exposure. Four weeks are short relative to the timescales relevant to mine-site remediation.
2. The microcosms are rich in Fe and sulphate; abiotic Fe(II) generated by microbial Fe(III) reduction could reduce U(VI). Current data cannot disentangle direct enzymatic reduction from indirect pathways.
3. Community changes are described qualitatively (e.g., enrichment of Propionibacteriaceae, Desulfobulbus). How many samples and replicates did you analyse?
4. The Introduction presents the work as broadly relevant, yet experiments use only one low uranium concentration at circum-neutral pH (7.5–8.0). Discuss applicability to a wider range of field conditions.
5. You appear to use indigenous microbes, but which organisms were present before the experiment? No analysis is provided.
6. Typographical issues: “electro-dense” should be “electron-dense”? “Crippled” nanoparticles seems odd; perhaps “aggregated” is better. Please check and revise the entire manuscript.

Reviewer #2

(Remarks to the Author)

This study reports the first observation of a stable U(V) species in a non-axenic microbial system, which is scientifically meaningful and contributes valuable insight into uranium biogeochemistry. However, the manuscript still requires further revision in terms of methodological transparency, data consistency, and scientific interpretation. The following questions are raised for the authors' consideration:

1. How were the uranium removal stages precisely controlled during the experiment? The authors mention sampling at different stages but do not clarify whether real-time monitoring or predefined reaction durations were used. In addition, the procedures used to separate solid particles from the suspension are not described. Were anaerobic conditions maintained during separation and preservation? These details are critical for ensuring the accuracy of subsequent spectroscopic characterizations.
2. Why did the authors use different amplitude reduction factors (S_{02}) in the EXAFS fitting of different samples? S_{02} , commonly ranging from 0.7–1.0, accounts for passive electron screening and should be determined consistently—typically based on standard references. Some of the values used fall outside this range. Were they treated as free-fitting parameters or fixed values? How do variations in S_{02} affect the derived coordination numbers? Clarification is needed regarding the rationale and consistency of these values.
3. Multiple bacterial strains were present in the experimental system, but the STEM analysis was performed on a single bacterial cell. Is this observation representative of the overall system? Have differences among strains in uranium reduction or nanoparticle formation been considered? Additionally, the reported 2–5 nm uranium nanoparticles are not accompanied by quantitative particle size distribution or edge-resolution data. Please justify the representativeness of the selected cell and provide supporting measurements if available.
4. There are inconsistencies in the number of significant figures throughout the manuscript. For instance, line 179 reports “3727 eV,” whereas line 183 reports “3726.5 eV.” It is recommended to adopt a consistent level of numerical precision across the manuscript to maintain scientific rigor and clarity.
5. Some terminologies and units are not used consistently or correctly throughout the manuscript. For instance, the redox potential is written as “EH” in uppercase, which is non-standard; it should be written as “Eh,” with only the first letter capitalized. Additionally, the unit for k-space in EXAFS analysis should be specified as \AA^{-1} (inverse ångström). The authors are advised to carefully review the entire manuscript, including the main text, figures, and captions, to ensure that all units, notations, and abbreviations conform to accepted scientific conventions and journal formatting standards.

Reviewer #3

(Remarks to the Author)

This study revealed the formation of pentavalent uranium species during bioremediation of uranium(VI)-contaminated mine water using glycerol as the electron donor. Through advanced analytical techniques—including HERFD-XANES, EXAFS, high-resolution transmission electron microscopy coupled with EDX and SAED, the oxidation state, speciation and

molecular structure of U(V)-containing compounds were characterized. Additionally, under aerobic conditions, long-term stability of pentavalent uranium (U(V)) species was observed, indicating U(V) species were resistant to re-oxidation compared with U(IV). Such findings can provide valuable insights for the bioremediation.

[1] The novelty of this study lies in the comprehensive characterization of U(V) species during bioremediation of uranium(VI)-contaminated mine water. Nevertheless, U(V) intermediates during the U(VI) reduction has been found in the established literature. Emphasis should be placed on the elucidating the mechanisms underlying U(V) formation, as this could enhance our understanding of the U(VI) reduction process.

[2] Caution is advised when interpreting 16S rRNA gene sequencing results. For example, the proposal of one electron transfer reaction remains speculative. Solid and direct evidence is needed to support such kind of claims.

[3] Overall, the experimental methodology and description were detailed and sound. However, classification of ASV and taxonomy annotation at 97% similarity is unclear (L564-565). Generally, OTUs (Operational Taxonomic Units) are often classified using a 97% similarity threshold. ASV is defined as unique DNA sequences.

[4] Although there is abiotic control, characterization of U(V) species within the abiotic control is not reported. A direct comparison between U(V) species in the abiotic and biotic conditions could provide deeper insight into the chemical versus microbial pathways of U(VI) reduction.

[5] L30, 16S rRNA "gene" sequencing.

[6] L113-120, the sentence is awkwardly long.

[7] The figures can be further reorganized and improved. For example, Figure 1 a-d could be stacked and compiled together in one figure to better illustrate differences under different conditions.

[8] What does B4 and B8 represent? Detailed information should be provided.

Reviewer #4

(Remarks to the Author)

The manuscript explores the microbial reduction of U(VI) to lower oxidation states in weakly U-contaminated mine water, using glycerol as an electron donor. The integration of advanced spectroscopic and microscopic techniques, such as HERFD-XANES, EXAFS, and HRTEM, provides valuable insights into uranium redox transformations. However, while the study presents novel observations, several critical issues limit its suitability for publication at this stage. Below are detailed comments and recommendations to improve the scientific robustness and clarity of the manuscript.

1. The manuscript presents identification of U(V)-carbonate complexes via EXAFS and U(V) in FeUO₄ nanoparticles via HRTEM. However, the authors note that discrepancies in sample preparation (wet paste for EXAFS vs. dehydrated material for HRTEM) may have impeded simultaneous detection of both species. To strengthen this conclusion, I recommend conducting parallel analyses on identically prepared samples or employing cryo-electron microscopy to preserve water-soluble complexes and minimize dehydration artifacts. This would provide stronger evidence for the coexistence of distinct U(V) species.

2. While 16S rRNA data show enrichment of taxa such as Propionibacteriaceae and Desulfobulbus following glycerol addition, their roles in U(VI) reduction remain speculative. The manuscript would benefit from the inclusion of functional gene analyses (e.g., *mtr*, *omc*, or other dissimilatory metal reduction genes), metatranscriptomic profiling, or pure culture experiments confirming U(VI) to U(V) reduction capabilities. These approaches would clarify causal links between microbial function and uranium redox transformation.

3. The claim that biogenic U(V) remains stable under oxic conditions for four weeks is novel, but the experimental parameters governing oxygen exposure are insufficiently described. Details such as oxygen concentration, gas exchange surface area, and aeration conditions should be clearly reported. Moreover, the increase in U(V) from 30% to 53% lacks statistical support—please provide replicate data, standard deviations, and statistical significance (e.g., p-values) to validate the reliability of these observations.

4. Uranium concentration decreased by 25-36% in the glycerol-free and sterilized controls, and the authors attribute this to biosorption. However, abiotic factors such as adsorption onto container surfaces or mineral particulates cannot be excluded. I recommend including additional blank controls (e.g., sterile mine water without biomass), or conducting kinetic biosorption experiments to quantify the role of biological vs. abiotic processes in uranium loss.

5. The manuscript suggests that the long-term stability of U(V) is mediated by FeU(V)O₄ nanoparticles or microbial biomass. However, similar stability of FeUO₄ under oxic conditions has already been reported (e.g., Crean et al., 2020). The authors should clearly delineate how their findings extend current knowledge such as through the use of weakly contaminated mine water or microbial community-driven U(V) formation while avoiding overstatement of previously established phenomena.

6. The HERFD-XANES peak assignments for U(VI) (~3727 eV), U(IV) (~3725 eV), and U(V) (~3726.5 eV) are appropriate. It would enhance clarity to briefly state whether these assignments were validated against reference standards or drawn from literature.

7. The identification of U(IV) as the dominant oxidation state, with concurrent detection of U(V), adds important nuance to our understanding of microbial uranium reduction. This aspect of the study is a clear strength.

8. The manuscript notes a delay in uranium reduction. Please elaborate on whether this could result from microbial acclimatization, strong carbonate complexation inhibiting reduction, or other chemical/biological factors.

9. The assignment of U(V) is a key conclusion. Please clarify whether this spectral feature was confirmed through comparison with known U(V) reference spectra or solely inferred from literature.

10. Given the well-known instability of U(V), a brief discussion on the potential stabilization mechanisms in your system—such as complexation with biogenic ligands or mineral surface association—would be informative.

11. Table S2 attributes uranium loss in control microcosms to biosorption. Please provide either references supporting this conclusion or additional data distinguishing biosorption from abiotic adsorption/reduction.

Version 2:

Reviewer comments:

Reviewer #1

(Remarks to the Author)

The authors addressed my concerns and comments. I agree to accept the paper's publication.

Reviewer #2

(Remarks to the Author)

The revised manuscript is improved, but several issues still need further clarification.

1. The TEM data lack panoramic images that include multiple cells. Although the authors state that five cells and 231 nanoparticles were examined, the manuscript and SI present only close-up TEM/HRTEM images from single or very limited cells. Without panoramic views showing multiple cells within the same field of view, the claim of consistent nanoparticle formation across cells is not adequately supported.

2. The use of uranium removal percentages (20%, 60%, 90%) to define reaction stages is conceptually misleading, as uranium reduction and the associated microbial and geochemical processes are inherently time-dependent. Reaction time should serve as the primary criterion for defining reaction stages.

3. The interpretation of the unusually low S_{σ^2} value (0.60) in the EXAFS fitting is not physically justified. S_{σ^2} should remain relatively constant for a given absorption edge. A refitting procedure with S_{σ^2} fixed to the reference value is recommended to test the robustness of the structural model.

Raw U L_{β} -edge spectra are not provided. Only processed HERFD-XANES and EXAFS data are presented, preventing independent evaluation of data quality and processing procedures. Raw spectral data should be included in the supplementary materials.

Minor Revision is therefore recommended before the manuscript can be further considered.

Reviewer #3

(Remarks to the Author)

The authors have revised the manuscript accordingly. I recommend publication after addressing the following minor concerns.

(1) Although the potential indirect abiotic reduction of U by biogenic hydrogen sulfide and/or Fe(II) cannot be ruled out, the authors did not measure hydrogen sulfide or Fe(II) species in their samples. Caution should therefore be exercised when discussing this process in the Conclusion. The Conclusion should also be more succinct and focused on the major findings.

(2) L518-521, While fermenters can generate electron donors from glycerol to support SRB activity, the authors previously noted that SRB can utilize glycerol directly (L410–413). This point should be clarified for consistency.

(3) L48-49, L669-671, grammatically incorrect.

(4) U(IV) at L123; "hexavalent U" "Tetravalent U" at L167. The U(IV), U(V) and U(VI) should be spelled out in full at their first appearance in the manuscript. Thereafter, the abbreviated forms may be used.

(5) Fig. S7 is not cited in the manuscript.

Reviewer #4

(Remarks to the Author)

The manuscript has been well revised and can be accepted for publication at current form.

Version 3:

Reviewer comments:

Reviewer #2

(Remarks to the Author)

The authors have carefully revised the manuscript. It is recommended for publication after the following minor revisions:

The manuscript suggests that mineral surface processes and Fe phases may contribute to U stabilization. Notably, beyond surface sorption/co-precipitation, U may also migrate into Fe-bearing clay interlayers and become persistently sequestered within confined microenvironments (doi: 10.1016/j.watres.2025.123582; 10.1016/j.gca.2009.07.002). The Discussion would benefit from a more detailed comparison of surface-driven mechanisms versus "interlayer confinement" processes, including their key distinctions and potential coupling, to better articulate the environmental relevance and practical implications of this study.

Reviewer #3

(Remarks to the Author)

The manuscript has been revised accordingly and can be accepted for publication.

Version 4:

Reviewer comments:

Reviewer #2

(Remarks to the Author)

The manuscript has been satisfactorily revised, and I recommend acceptance.

Response to Reviewers and Editor:

This document contains our point-by-point responses to the comments from the editor and the reviewers regarding the manuscript “*Pentavalent and Tetravalent Uranium Formation via Glycerol-Stimulated Bacteria in Mine Water*” (NCOMMS-25-25237A-Z)

In the first instance, we would like to thank the editor and reviewers for their constructive comments and suggestions, which greatly helped us to improve the clarity and impact of our manuscript.

In accordance with the editorial guidelines, all changes introduced in the revised version are highlighted in **blue font** for clarity.

Additionally, we have added Dr. Sean Ting-Shyang Wei as a co-author in the revised submission, based on his contribution to new data analyses incorporated in response to the reviewers’ requests. The corresponding *author change form* has been submitted accordingly.

Reviewer #1 (Remarks to the Author):

The manuscript presents a combination of advanced spectroscopic, microscopic, and geochemical techniques to demonstrate that biological U(VI) reduction proceeds through, and partially terminates in, a pentavalent intermediate (FeU(V)O_4) that survives for weeks under oxic, near-neutral conditions. Nevertheless, the significance is not well articulated and many aspects of microbiology, mechanistic interpretation, and data presentation require substantial strengthening. Particularly, the microbial analysis should be improved significantly. Regardless, I still think, this observation is quite novel and potentially impactful for uranium remediation science. I therefore recommend major revision. If the authors can significantly improve the manuscript, then I will reconsider.

Specific comments:

- 1. Line 26, Abstract: In the first sentence I do not see the significance or importance of the issue.**

We have revised the beginning the Abstract to emphasise the global importance of uranium (U) contamination and the need to better understand its biogeochemistry.

[Revised Abstract, lines 28–29]

- 2. Line 30: Is 16S rRNA analysis sufficient? Please justify.**

We agree that 16S rRNA sequencing provides only taxonomic context. Our conclusions on U(V)/U(IV) formation rely exclusively on spectroscopic and microscopic evidences. The microbial analysis is presented to contextualise the observed geochemical changes, and we now make this explicit in the main text. To further strengthen this point, we added metatranscriptomic baseline data (time 0) showing active sulphur metabolism, consistent with the observed community shifts after glycerol addition.

[Revised Result and Discussion, lines 416–427; Methods, lines 666–696; Supplementary Material, Table S6, line 209 and Figure S11, line 169]

- 3. Line 36: It is better to add “To the best of our knowledge” before “first report.”** The Abstract has been rewritten during revision, and the specific phrasing “first report” has been removed.

[Revised Abstract, lines 28–41]

- 4. Introduction: The structure is quite messy. Lines 52–65 devote too much space to the situation in Germany while offering only a few words on the global context. What about the international situation?**

The Introduction has been revised to first outline U contamination as a global issue, with examples from different regions, before narrowing down to Europe and finally Germany. The Schlema-Alberoda site is then introduced as a representative post-mining case relevant to our experiments.

[Revised Introduction, lines 45–58]

- 5. Lines 66–68: Which physicochemical methods were used, and why did they fail? Provide comparative evidence; simply stating that bioremediation is effective is overly assertive.**

We clarified that the conventional treatment in Schlema-Alberoda consists of a modified lime precipitation process, which has effectively reduced contaminant loads over three decades. However, this method is cost-intensive, generates large volumes of secondary sludge, and requires continuous operation until natural dilution sufficiently decreases contaminant concentrations. Therefore, *in-situ* immobilisation strategies such as bioremediation are being investigated to accelerate U concentration decline and shorten treatment duration. We have revised the introduction accordingly and added comparative evidence from field studies highlighting the site-dependent performance of bioremediation.

[Revised Introduction, lines 62–74; lines 75–81]

- 6. Line 72: Clarify whether biomineralisation includes enzymatic reduction. The mechanism of uranium bioreduction is not explained in sufficient depth.**

We have revised the Introduction to clearly distinguish between biomineralisation and bioreduction.

[Revised Introduction, lines 81–87]

- 7. Lines 75–79: The logic is difficult to follow. The definition of waste glycerol is vague. Why choose glycerol when other waste products can serve as carbon sources? Is it more efficient? Is there large-scale waste glycerol production? Have you calculated the cost, including collection and transportation, compared with commercial carbon sources? Is carbon supply the main cost driver in bioremediation? This is unclear.**

We clarified that conventional electron donors such as acetate or lactate are suitable at laboratory scale but economically unsustainable for field applications. In contrast, biodiesel production generates large quantities of crude glycerol (~10% of total output), an impure and low-value by-product that is regionally available (e.g., VERBIO Bitterfeld GmbH, Saxony-Anhalt, Germany). This makes it a practical and cost-effective carbon source for biostimulation under site-specific conditions. In our system, glycerol demonstrated higher effectiveness than other tested electron donors (gluconic acid and vanillic acid; Newman-Portela et al., 2024, *Environ. Sci. Pollut. Res.*). While no full economic analysis was conducted, we note that carbon supply is a major cost component in field-scale bioremediation. At the Schlema-Alberoda site alone, more than €1 billion have been invested in remediation activities to date (Wismut GmbH, 2016), underscoring the scale and long-term operational costs of water treatment. Given the large and well-mixed mine-water body ($\approx 7 \times 10^6$ m³), sustainable complementary strategies such as biostimulation with residual glycerol could help reduce operational demands and support the long-term management of U-contaminated mine waters.

[Revised Introduction, lines 89–99]

- 8. Lines 80–89: The first paragraph already states that U(V) forms and is often overlooked during reduction from U(VI) to U(IV). Repeating this here is redundant; consider merging.**

We have merged the two paragraphs into a single, concise section that introduces U(V), summarises evidence for its role as an intermediate, and highlights its potential stability under environmental conditions. This revision removes redundancy and improves the focus of the Introduction.

[Revised Introduction, lines 100–109]

- 9. Lines 97–98: The mere absence of previous studies does not automatically establish importance; explain why this gap matters.**

We agree that the mere absence of previous studies alone does not establish significance. We have therefore revised the text to explain why this knowledge gap is critical. Specifically, previous reports of U(V) formation have been limited to pure bacterial cultures or to systems with artificially high U(VI) concentrations under chemical simplified conditions, which do not represent post-mining waters. Demonstrating U(V) formation in low-U, heterogeneous microbial communities is essential, as these are the systems that ultimately control U mobility and long-term remediation outcomes.

[Revised Introduction, lines 112–118]

- 10. Lines 100–108: Again redundant; please streamline.**

We agree and have merged this section removing repetition and presenting a single, concise paragraph that introduces U(V), summarises experimental and natural evidence, and outlines stabilisation mechanisms.

[Revised Introduction, lines 100–109]

- 11. Lines 109–129: State the study's aim succinctly and concisely outline the research content rather than listing specific methods and instruments. Indicate the perspectives you address, how you meet your aim, and the overall significance (which is currently unclear).**

We agree and have rewritten this section to state the aim clearly, summarise the approach in general terms, and highlight the main findings and their significance.

[Revised Introduction, lines 119–128]

- 12. Results and Discussion, Line 151: What is the dash for? Please clarify.**

The dash was a formatting error in the submitted version. We have corrected it in the revised manuscript.

[Revised Result and Discussion, lines 151–152]

- 13. Figure 3: Explain how you calculated the mass balance of uranium. Do the three oxidation states sum exactly to 100 %?**

The U(IV), U(V), and U(VI) fractions were obtained using Iterative Target Factor Analysis (ITFA) of U M₄-edge HERFD-XANES spectra. ITFA differs from a simple Linear Combination Fit (LCF) because it recalculates the spectral components, which can lead to small deviations from 100%. In our case, sums ranged from 103–107%, well within the estimated uncertainties ($\pm 5\%$ for U(IV), $\pm 10\%$ for U(V) and U(VI)). For clarity, we renormalised the fractions to 100% in Figure 1, and we have now made this explicit in the figure legend and Methods.

[Revised Result and Discussion, lines 202–203; Supplementary Information 2, lines 69–75]

- 14. Several paragraphs are overly long, making it easy for readers to lose interest; consider breaking them into shorter, coherent units.**

In the revised manuscript, long paragraphs have been divided into shorter, coherent units to improve readability and preserve the logical flow of the narrative.

- 15. The spectroscopic and microscopic results are very interesting, but the subtitle promises an integrated discussion that is not delivered. Strengthen the overall synthesis of results and discussion.**

We agree and have improved the integration of spectroscopy and microscopy and a concise summary to strengthen the synthesis.

[Revised Result and Discussion, lines 341–385]

- 16. My major concern: You present strong physicochemical data, yet for the microbial aspect you rely only on 16S rDNA. This technique mainly provides relative abundance and offers limited functional insight. You need greater depth to explain why and how microbes form U(V). Consider adding metagenomics, transcriptomics, or quantitative PCR.**

We recognise that 16S rRNA sequencing primarily offers a taxonomic overview and does not resolve functional pathways. Our conclusions on U(V)/U(IV) formation rely exclusively on spectroscopic and microscopic evidences. The microbial analysis is presented to contextualise the observed geochemical changes, and we now make this explicit in the main text. To further strengthen this point, we added Metatranscriptomic baseline data (time 0) showing active sulphur metabolism, consistent with the observed community shifts after glycerol addition.

[Revised Result and Discussion, lines 416–427; Methods, lines 666–696; Supplementary Material, Table S6, line 209 and Figure S11, line 169]

- 17. Lines 373–381: Fermentative bacteria may not necessarily be related to uranium remediation; please clarify their relevance.**

We have clarified in the revised text that fermentative bacteria may contribute indirectly to U reduction by producing metabolites that serve as electron donors for sulphate- and metal-reducing bacteria. These latter groups are key groups for U(VI) reduction, and fermenters sustain the trophic interactions that allow them to thrive.

[Revised Result and Discussion, lines 405–409; 442–450 and 518–521]

- 18. Abrupt mention of SRB: Introducing sulphate-reducing bacteria here feels abrupt. Provide background for readers and mention their relevance earlier in the Introduction.**

We agree that the introduction of sulphate-reducing bacteria was abrupt. In the revised Introduction, we now provide background on the main microbial groups involved in U(VI) reduction, including SRB, to ensure a smoother transition.

[Revised Introduction, lines 87–90]

- 19. Lines 382–388: These are well-established facts; condense or cite authoritative reviews.**

We have condensed the paragraph to briefly note the dominance of *Desulfobulbus* and *Desulfovibrio* and their role in coupling glycerol oxidation to sulphate reduction. Authoritative papers are cited to support these well-established characteristics.

[Revised Result and Discussion, lines 410–413]

- 20. Lines 395–396: Coupled S, Fe, and U biogeochemical cycles are a vast topic; omit this unless you treat it in detail.**

We agree and have removed the reference to coupled cycles. The paragraph now ends with a focused description of the microbial taxa enriched in our microcosms.

- 21. I do not see a thorough discussion linking microbial results with the earlier physicochemical findings; please integrate these aspects more deeply.**

We have revised the Discussion to explicitly integrate microbial, geochemical, and spectroscopic results. The revised text links fermentative activity with sulphate and Fe

reduction and the associated E_h decrease, creating conditions for U(VI) reduction observed by HERFD-XANES, EXAFS and HRTEM.

[Revised Result and Discussion, lines 442–460]

- 22. Lines 419–445: The manuscript proposes U(V) formation, but its stability is discussed mainly with reference to literature. More importantly, the study’s aim is still unclear. While the formation of U(V) is intriguing, what is its broader significance? You note that U(V) is more stable than expected and may aid U(VI) bioremediation, but the mechanism needs explanation. Clarify this for readers.**

In the original version, the aim and broader significance of our study were not sufficiently clear. We have therefore restructured the text to state consistently that the aim of this study is to investigate the glycerol based biostimulation of microbes involved in reduction of U(VI) in mine waters and to determine whether U(V) forms and persists under environmentally relevant conditions, as well as to assess its potential role in U immobilisation for bioremediation.

Our findings show that U(V), usually considered a transient intermediate, persisted for 130 days under anoxic conditions and for four weeks under oxic exposure. This expands the conventional U(VI)/U(IV) framework and suggests that U(V) may act as a buffer state, delaying U(VI) remobilisation when U(IV) is re-oxidised.

Although the precise stabilisation mechanisms remain to be fully evaluated, our data and the literature point to microbial biomass, organic ligands, and Fe-containing nanoparticles as contributing factors.

[Revised Introduction, lines 119–128; Result and Discussion, lines 442–460 and lines 469–504]

Other concerns and comments:

- 1. The key advance is the persistence of U(V) after four weeks of air exposure. Four weeks are short relative to the timescales relevant to mine-site remediation.**

We acknowledge that four weeks is short compared with the decades relevant for remediation. However, this result represents an important advance, providing the first direct evidence that U(V) can persist under environmentally realistic conditions, including low U concentrations, complex microbial communities, and oxic–anoxic transitions, as resolved by advanced spectroscopic and microscopic analyses. While this does not yet demonstrate stability over decades, it challenges the assumption that U(V) is intrinsically transient and indicates that longer-term persistence is plausible.

- 2. The microcosms are rich in Fe and sulphate; abiotic Fe(II) generated by microbial Fe(III) reduction could reduce U(VI). Current data cannot disentangle direct enzymatic reduction from indirect pathways.**

We have revised the Discussion to explicitly address this limitation. The text now states that both direct enzymatic reduction and indirect abiotic pathways (via Fe(II) or sulphides) are likely operating, but their relative contributions cannot be resolved with the current dataset.

[Revised Result and Discussion, lines 155–156; 432–434 and 447–450]

- 3. Community changes are described qualitatively (e.g., enrichment of Propionibacteriaceae, Desulfobulbus). How many samples and replicates did you analyse?**

We analysed microbial communities at the end of the 130-day incubation in two representative glycerol-amended microcosms (B4 and B8). Each was processed with three independent replicates ($n = 3$). Sequencing results were consistent among replicates, and are

shown in Supplementary Figure S7 and Table S3. We have clarified this in the Methods and in the Supplementary Figure legend.

[Revised Methods, lines 641–664; Revised Supplementary Material, Figure S10 line 162 and Table S5 line 198]

- 4. The Introduction presents the work as broadly relevant, yet experiments use only one low uranium concentration at circum-neutral pH (7.5–8.0). Discuss applicability to a wider range of field conditions.**

Our experiments were designed to simulate realistic post-mining conditions at the Schlemma-Alberoda site. These included neutral to alkaline pH, high carbonate alkalinity, and approximately 1 mg/L U. Such conditions are representative of many mining-impacted waters worldwide (Gandhi et al., 2022, *Sci. Total Environ.*). We acknowledge that other sites may differ in pH, U speciation, or the availability of electron acceptors. These differences are important because they can influence the relative proportions of U(V) and U(IV). Variations in Fe mineralogy, carbonate content, or the presence of organic ligands may also stabilise U(V) to different extents. Although our study does not cover the full range of possible conditions, the pathway we describe involves coupled biotic and abiotic U(VI) reduction that yields U(IV) and stabilised U(V). This mechanism is likely to occur at other sites as well. The persistence of U(V) under the common scenario of neutral, carbonate-rich waters with low U concentrations is therefore highly relevant. It provides key insights into U behaviour beyond the traditional U(VI)/U(IV) paradigm. It also informs the design of bioremediation strategies that can be adapted to specific local geochemical settings.

[Revised Conclusions, lines 525–543]

- 5. You appear to use indigenous microbes, but which organisms were present before the experiment? No analysis is provided.**

The microcosms contained only indigenous microbes, as no allochthonous inoculum was added. Their baseline composition had already been characterised in our previous study (Newman-Portela et al., 2024, *Environ. Sci. Pollut. Res.*), which reported dominant phyla including Proteobacteria, Campylobacterota, Patescibacteria, Verrucomicrobiota, and Nitrospirota, with functional groups such as nitrate reducers, sulphur oxidisers, potential sulphate reducers, iron oxidisers, and potential iron reducers. In the present study we therefore focused on the shifts induced by glycerol biostimulation.

[Revised Result and Discussion, lines 388-396]

- 6. Typographical issues: “electro-dense” should be “electron-dense”? “Crippled” nanoparticles seems odd; perhaps “aggregated” is better. Please check and revise the entire manuscript.**

The terms have been corrected to “electron-dense” and “aggregated nanoparticles,” and the entire manuscript has been carefully revised for consistency and typographical accuracy.

[Results, lines 281, 303, 308 and 312]

Reviewer #2 (Remarks to the Author):

This study reports the first observation of a stable U(V) species in a non-axenic microbial system, which is scientifically meaningful and contributes valuable insight into U biogeochemistry. However, the manuscript still requires further revision in terms of methodological transparency, data consistency, and scientific interpretation.

The following questions are raised for the authors' consideration:

- 1. How were the uranium removal stages precisely controlled during the experiment? The authors mention sampling at different stages but do not clarify whether real-time**

monitoring or predefined reaction durations were used. In addition, the procedures used to separate solid particles from the suspension are not described. Were anaerobic conditions maintained during separation and preservation? These details are critical for ensuring the accuracy of subsequent spectroscopic characterizations.

U removal stages were not predefined by incubation time but determined based on real-time ICP-MS measurements of dissolved U concentration. Sampling points corresponded to ~20%, ~60%, and ~90% decreases in U concentration (Supplementary Fig. S2–S3), ensuring that spectroscopic samples represented distinct bioreduction stages. Black precipitates were separated by centrifugation (4,020 × g, 15 min, Hettich EBA 21, Germany) inside an anaerobic glove box, and pellets were cryogenically frozen to preserve redox state. HERFD-XANES samples were loaded as wet pastes in Kapton-sealed holders, while EXAFS samples were frozen in polyethylene holders and measured under cryogenic conditions. These details have been added to the Methods section to ensure transparency and demonstrate that spectroscopic results reliably reflect U speciation formed in the microcosms.

[Revised Methods, lines 563–568]

- 2. Why did the authors use different amplitude reduction factors (S02) in the EXAFS fitting of different samples? S02, commonly ranging from 0.7–1.0, accounts for passive electron screening and should be determined consistently—typically based on standard references. Some of the values used fall outside this range. Were they treated as free-fitting parameters or fixed values? How do variations in S02 affect the derived coordination numbers? Clarification is needed regarding the rationale and consistency of these values.**

The only pure references with known structures are $\text{UO}_2(\text{CO}_3)_3^{4-}$ and crystalline uraninite (UO_2), for which S_0^2 could be determined precisely. However, they are not necessarily the best references for the other species, since these differ structurally and occur in different physical states (wet paste vs. aqueous solution). All spectra were re-fitted, and the determination of S_0^2 was constrained using well-justified structural parameters, specifically, a fixed average coordination number $\text{CN}(\text{O}_{\text{ax}}) = 2$ for the carbonate U(V/VI) species and $\text{CN}(\text{O}) = 8$ for the first-shell oxygen atoms of the uraninite species. After determining S_0^2 , these fixed parameters were released for the final fits, as presented in Table 1. The notably lower S_0^2 value of 0.6, observed for the extracted spectrum of the mixed U(V/VI) species, can be explained by the reduction of EXAFS amplitude due to destructive interference between the signals of the two coexisting species.

[Revised Result and Discussion, lines 250–255; Table 1, line 242; Figure 2 line 234]

- 3. Multiple bacterial strains were present in the experimental system, but the STEM analysis was performed on a single bacterial cell. Is this observation representative of the overall system? Have differences among strains in uranium reduction or nanoparticle formation been considered? Additionally, the reported 2–5 nm uranium nanoparticles are not accompanied by quantitative particle size distribution or edge-resolution data. Please justify the representativeness of the selected cell and provide supporting measurements if available.**

To ensure representativeness, STEM/HRTEM/SAED analyses were performed across five different bacterial cells. In total, 231 U nanoparticles were identified and classified as $\text{FeU}^{(\text{V})}\text{O}_4$ (55.4%), uraninite (40.3%), and pyrite (4.3%) (Fig. S8; Table S3). In addition, TEM-based measurements of 160 nanoparticles were used to calculate the Equivalent Circular Diameter (ECD), confirming that most ranged between 2 and 3 nm, with a minority of larger particles extending the distribution tail (Fig. S9; Table S4). These analyses

demonstrate that the observations are not limited to a single cell but reflect a consistent pattern across the system. We have revised the Results and Supplementary Information to clarify this point and to include quantitative size distribution data.

[Revised Result and Discussion, lines 334–340, Methods, 634–640; Supplementary Information 3, lines 84–101; Supplementary Material, Figure S8 line 146, Figure S9 line 153; Table S3 line 185 and Table S4 line 190]

- 4. There are inconsistencies in the number of significant figures throughout the manuscript. For instance, line 179 reports “3727 eV,” whereas line 183 reports “3726.5 eV.” It is recommended to adopt a consistent level of numerical precision across the manuscript to maintain scientific rigor and clarity.**

We have carefully revised the manuscript and Supplementary Information to standardise numerical precision according to the resolution of each technique. All reported maxima are now consistently given with one decimal place.

- 5. Some terminologies and units are not used consistently or correctly throughout the manuscript. For instance, the redox potential is written as “EH” in uppercase, which is non-standard; it should be written as “Eh,” with only the first letter capitalized. Additionally, the unit for k-space in EXAFS analysis should be specified as Å⁻¹ (inverse ångström). The authors are advised to carefully review the entire manuscript, including the main text, figures, and captions, to ensure that all units, notations, and abbreviations conform to accepted scientific conventions and journal formatting standards.**

We have revised the entire manuscript, figures, and captions to ensure consistent use of scientific terminology and units.

[Revised Result and Discussion, lines 237 and 238; Table 1 line 242; lines 250–255, line 266, 272 and 273]

Reviewer #3 (Remarks to the Author):

This study revealed the formation of pentavalent uranium species during bioremediation of uranium(VI)-contaminated mine water using glycerol as the electron donor. Through advanced analytical techniques-including HERFD-XANES, EXAFS, high-resolution transmission electron microscopy coupled with EDX and SAED, the oxidation state, speciation and molecular structure of U(V)-containing compounds were characterized. Additionally, under aerobic conditions, long-term stability of pentavalent uranium (U(V)) species was observed, indicating U(V) species were resistant to re-oxidation compared with U(IV). Such findings can provide valuable insights for the bioremediation.

- 1. The novelty of this study lies in the comprehensive characterization of U(V) species during bioremediation of uranium(VI)-contaminated mine water. Nevertheless, U(V) intermediates during the U(VI) reduction has been found in the established literature. Emphasis should be placed on the elucidating the mechanisms underlying U(V) formation, as this could enhance our understanding of the U(VI) reduction process.**

We agree, U(V) has indeed been reported as an intermediate, but predominantly under controlled laboratory conditions with pure microbial cultures. The novelty of our work does not lie in the first observation of U(V), but in demonstrating of its biogenic formation and performing an integrated, multi-technique characterisation under realistic mine-water conditions, featuring high carbonates content, low U concentration (~1 mg L⁻¹), and an indigenous mixed microbial community. Most notably, we show that U(V), rather than

acting solely as a transient intermediate, can persist under oxic conditions as a stabilised species.

We demonstrate the coexistence of U(IV) as biogenic uraninite and U(V) as aqueous carbonate complexes and $\text{FeU}^{(\text{V})}\text{O}_4$ nanoparticles. These species persisted for 130 days under anoxic conditions and four weeks after oxic exposure, providing the first direct evidence of stable U(V) in such systems.

Although a full mechanistic resolution remains beyond the current analytical capabilities, the combined geochemical, microbiological, spectroscopic, and microscopic evidence consistently indicates coupled biotic–abiotic processes. The observed patterns are coherent with an initial enzymatic one-electron transfer during microbial U(VI) reduction, followed by abiotic stabilisation through Fe-bearing nanoparticles and organic ligands. These data thus support a plausible and internally consistent mechanistic interpretation, rather than mere speculation. Importantly, U(V) persistence represents a distinct and credible pathway for uranium immobilisation that extends beyond the conventional U(VI)/U(IV) paradigm and may be decisive for achieving regulatory remediation endpoints. Other processes, including indirect reduction by biogenic Fe(II) and hydrogen sulphides, may also contribute.

[Revised Result and Discussion, lines 442–460 and lines 480–489]

- 2. Caution is advised when interpreting 16S rRNA gene sequencing results. For example, the proposal of one electron transfer reaction remains speculative. Solid and direct evidence is needed to support such kind of claims.**

We agree that 16S rRNA sequencing does not provide direct evidence of molecular mechanisms. In our manuscript, references to electron transfer are explicitly based on previous studies with model pure bacterial cultures. For clarity, we emphasise that the proposed electron transfer mechanism derives from prior literature and is not demonstrated by our data, but is mentioned only as a possible pathway (Molinas et al., 2021, *Environ. Sci. Technol.*). Here, the 16S rRNA gene sequencing data are used solely to contextualise microbial community shifts after glycerol amendment, and not as direct functional proof. To provide further context, we have included baseline metatranscriptomic data from Schlemma-Alberoda mine water in the Supplementary Information, which confirm expression of genes for sulphate reduction. This supports the geochemical interpretation but does not constitute mechanistic evidence.

[Revised Result and Discussion, lines 416–427; Methods, lines 666–696; Supplementary Material, Table S6, line 209 and Figure S11, line 169]

- 3. Overall, the experimental methodology and description were detailed and sound. However, classification of ASV and taxonomy annotation at 97% similarity is unclear (L564-565). Generally, OTUs (Operational Taxonomic Units) are often classified using a 97% similarity threshold. ASV is defined as unique DNA sequences.**

In the revised version we clarify that reads were processed with QIIME2 (v2024.10) and DADA2 to infer amplicon sequence variants (ASVs) directly, without clustering into OTUs. Taxonomy was assigned using a Naive Bayes classifier trained on the SILVA 138 reference database, which is clustered at 99% similarity. Therefore, the 97% threshold was not applied to our dataset.

[Revised Methods, lines 659–664]

- 4. Although there is abiotic control, characterization of U(V) species within the abiotic control is not reported. A direct comparison between U(V) species in the abiotic and biotic conditions could provide deeper insight into the chemical versus microbial pathways of U(VI) reduction.**

We agree that a direct comparison of U(V) species in abiotic and biotic microcosms would provide valuable insights into chemical versus microbial pathways. However, in our abiotic controls the residual U concentrations after 130 days were below the thresholds for reliable HERFD-XANES/EXAFS analysis, preventing a robust assignment of U(V). Redox monitoring also showed no sustained decrease in E_h or any evidence of U(VI) reduction, so we conservatively attribute the partial U loss to abiotic sorption. By contrast, glycerol-amended microcosms produced clear U(V)/U(IV) signatures, consistent with microbial activity. We agree that dedicated spectroscopic characterisation of abiotic controls, designed to keep U concentrations within a measurable range, would be valuable.

5. L30, 16S rRNA “gene” sequencing.

We now use the precise term “16S rRNA gene sequencing” consistently throughout the manuscript and Supplementary Information and Material.

6. L113-120, the sentence is awkwardly long.

The sentence in lines 113–120 has been removed during revision, and the text was rewritten throughout to avoid overly long sentences while preserving the scientific content.

7. The figures can be further reorganized and improved. For example, Figure 1 a-d could be stacked and compiled together in one figure to better illustrate differences under different conditions.

We carefully considered the option of compiling the spectra into a single stacked figure. However, in the revised manuscript we decided to keep the spectra displayed separately. This layout enhances the visibility of the characteristic U(V) peak, which is a key strength of our dataset. In many previous studies, the U(V) feature is difficult to resolve, whereas in our experiments it is distinctly observed. The current arrangement was therefore chosen to highlight this feature and to ensure that the differences among the conditions remain easy to follow.

8. What does B4 and B8 represent? Detailed information should be provided.

B4 and B8 are two representative glycerol-amended microcosms with similar geochemical conditions sampled at the end of the 130-day incubation, when >90% of U(VI) had been removed. For each, three replicates ($n = 3$) were analysed. DNA was extracted from independent filter sections, pooled, and processed for 16S rRNA sequencing as described in the Methods. We have clarified this in the revised manuscript and in the legend of Supplementary Figure S10.

[Revised Methods, lines 644–649; Revised Supplementary Material, Figure S10 line 162 and Table S5 line 198]

Reviewer #4 (Remarks to the Author):

The manuscript explores the microbial reduction of U(VI) to lower oxidation states in weakly U-contaminated mine water, using glycerol as an electron donor. The integration of advanced spectroscopic and microscopic techniques, such as HERFD-XANES, EXAFS, and HRTEM, provides valuable insights into uranium redox transformations. However, while the study presents novel observations, several critical issues limit its suitability for publication at this stage. Below are detailed comments and recommendations to improve the scientific robustness and clarity of the manuscript.

1. The manuscript presents identification of U(V)-carbonate complexes via EXAFS and U(V) in FeUO₄ nanoparticles via HRTEM. However, the authors note that discrepancies in sample preparation (wet paste for EXAFS vs. dehydrated material for HRTEM) may have impeded simultaneous detection of both species. To strengthen this

conclusion, I recommend conducting parallel analyses on identically prepared samples or employing cryo-electron microscopy to preserve water-soluble complexes and minimize dehydration artifacts. This would provide stronger evidence for the coexistence of distinct U(V) species.

In our study, EXAFS was performed on wet pastes, which retain the aqueous or hydrated environments, maintaining water-soluble U species, allowing the detection of U(V)-carbonate complexes and uraninite. In contrast, HRTEM was applied to dehydrated thin sections, enabling the detection of crystalline nanoparticles such as $\text{FeU}^{(\text{V})}\text{O}_4$ and uraninite. The differences also reflect the complementary sensitivities of the two techniques. EXAFS provides a bulk average signal of the sample and is sensitive to dominant species, while HRTEM is a localised method that primarily resolves crystalline solids and offers limited ability to detect dissolved or amorphous species. All measurements were performed on the same bulk material and time point. Together, the results support the coexistence of aqueous U(V)-carbonate complexes and solid U(V)-bearing phases within the same microbial system. Parallel analyses with identical preparation or cryo-TEM could further strengthen this conclusion. However, it should be noted that uranyl-carbonate complexes are soluble and amorphous, and thus would be difficult, if not impossible, to detect them by TEM, even under cryogenic conditions. This limitation has been discussed in the literature, as cryo-TEM is typically not suitable for directly determining soluble actinide-carbonate complexes. Instead, such species are usually identified and quantified by spectroscopic techniques, including X-ray absorption spectroscopy (XAS, e.g., EXAFS and HERFD-XANES), UV-Vis absorption spectroscopy, and cryo time-resolved laser-induced fluorescence spectroscopy (cryo-TRLFS), which can resolve U oxidation state and coordination environment in systems (Monnier et al., 2015, *AIMS Biophys.*; Krawczyk-Bärsch et al., 2018, *J. Hazard. Mater.*; Kvashnina and Butorin, 2022, *Chem. Commun.*).

- 2. While 16S rRNA data show enrichment of taxa such as Propionibacteriaceae and Desulfobulbus following glycerol addition, their roles in U(VI) reduction remain speculative. The manuscript would benefit from the inclusion of functional gene analyses (e.g., *mtr*, *omc*, or other dissimilatory metal reduction genes), metatranscriptomic profiling, or pure culture experiments confirming U(VI) to U(V) reduction capabilities. These approaches would clarify causal links between microbial function and uranium redox transformation.**

We agree that 16S rRNA gene sequencing provides taxonomic resolution but does not, by itself, establish functional roles in U(VI) reduction. In our study, microbial community analysis was used to contextualize the geochemical, spectroscopic, and microscopic evidence for U(V)/U(IV) formation, rather than to directly infer specific enzymatic mechanisms. To address this more clearly, we have revised the text to emphasise that fermentative taxa likely contributed indirectly by generating electron donors that stimulated sulphate- and metal-reducing bacteria, which are more directly implicated in U(VI) reduction according to previous literature. Furthermore, we now report baseline metatranscriptomic data (time zero), which reveal expression of key genes in sulphate reduction pathways, supporting the presence of an active functional potential consistent with the observed biogeochemical shifts. While targeted functional gene assays or pure culture experiments would indeed strengthen causal inference, they were beyond the scope of this study. Importantly, the multiple independent lines of evidence presented (geochemical, spectroscopic, microscopic, and community-level data) provide robust support for U(V)/U(IV) formation under environmentally realistic conditions.

[Revised Result and Discussion, lines 416–427, lines 442–460, lines 480–504; Methods, lines 665–696; Supplementary Material, Table S6, line 209 and Figure S11, line 169]

- 3. The claim that biogenic U(V) remains stable under oxic conditions for four weeks is novel, but the experimental parameters governing oxygen exposure are insufficiently described. Details such as oxygen concentration, gas exchange surface area, and aeration conditions should be clearly reported. Moreover, the increase in U(V) from 30% to 53% lacks statistical support—please provide replicate data, standard deviations, and statistical significance (e.g., p-values) to validate the reliability of these observations.**

We have added details of the oxic stability test in the Methods section. The sample obtained at 90% U removal was exposed to ambient air in a closed laboratory (21 ± 1 °C) for 4 weeks, without active aeration and with protection against particulate contamination. After exposure, the sample was sealed for HERFD-XANES analysis.

Regarding reproducibility, multiple spectra were collected at different positions of the same sample (4–10 spectra, averaged), ensuring spatial consistency. The U(IV), U(V), and U(VI) fractions were quantified by Iterative Target Factor Analysis (ITFA) of U M₄-edge HERFD-XANES spectra. As ITFA recalculates spectral components, the sums initially ranged from 103–107%, which is within the expected uncertainties ($\pm 5\%$ for U(IV) and $\pm 10\%$ for U(V)/U(VI)). For clarity, the fractions were renormalised to 100% in Figure 1, and this is now explicitly stated in the legend and Methods.

Finally, we have softened the interpretation. Rather than describing an increase, we now emphasise the persistence of U(V) in substantial proportions after 4 weeks under oxic conditions, with redistribution among U(IV), U(V), and U(VI) consistent with partial re-oxidation of U(IV).

[Revised Result and Discussion, lines 202–203; Supplementary Information 2, lines 69–75]

- 4. Uranium concentration decreased by 25–36% in the glycerol-free and sterilized controls, and the authors attribute this to biosorption. However, abiotic factors such as adsorption onto container surfaces or mineral particulates cannot be excluded. I recommend including additional blank controls (e.g., sterile mine water without biomass), or conducting kinetic biosorption experiments to quantify the role of biological vs. abiotic processes in uranium loss.**

We appreciate this suggestion. In our study, the control microcosms included unamended mine water (SAC) and sterilized mine water with glycerol (ASA+G), as described in the Materials and Methods section. The sterilization step already suppresses microbial activity and therefore provides an abiotic reference. To address this, we have revised the text to conservatively describe the observed uranium decrease as partly due to abiotic/adsorptive removal. Moreover, no interactions between uranium and glycerol have been reported based on our previous studies (Newman-Portela et al., 2024, *Environ. Sci. Pollut. Res.*), supporting the suitability of ASA+G as an abiotic control.

[Revised Results and Discussion, lines 139–147; Methods, lines 550–554]

- 5. The manuscript suggests that the long-term stability of U(V) is mediated by FeU(V)O₄ nanoparticles or microbial biomass. However, similar stability of FeUO₄ under oxic conditions has already been reported (e.g., Crean et al., 2020). The authors should clearly delineate how their findings extend current knowledge such as through the use of weakly contaminated mine water or microbial community-driven U(V) formation—while avoiding overstatement of previously established phenomena.**

We acknowledge that the stability of FeU^(V)O₄ under oxic conditions has been reported previously. To avoid overstatement, we have revised the text to emphasise that our study

extends current knowledge by demonstrating the direct microbial formation of U(V) in weakly contaminated mine water. This contrasts with earlier work on synthetic $\text{FeU}^{(\text{V})}\text{O}_4$ particles produced under high-temperature synthesis conditions (Crean et al., 2020, *Environ. Sci.: Processes Impacts*).

In addition, we show the coexistence of aqueous U(V)-carbonate complexes and $\text{FeU}^{(\text{V})}\text{O}_4$ nanoparticles within the same microbial system. These findings highlight that U(V) stabilisation can occur under environmentally realistic conditions mediated by natural microbial communities. Such processes may influence U immobilisation relevant to long-term bioremediation.

[Revised Results and Discussion, lines 462–504]

- 6. The HERFD-XANES peak assignments for U(VI) (~3727 eV), U(IV) (~3725 eV), and U(V) (~3726.5 eV) are appropriate. It would enhance clarity to briefly state whether these assignments were validated against reference standards or drawn from literature.**

The HERFD-XANES peak assignments were validated using both reference standards and literature data. Reference spectra for U(IV) (UO_2) and U(VI) (uranyl nitrate) were measured during the same beamtime and used for energy calibration. The U(V) assignment was validated against the UMoO_5 reference spectrum (Pan et al., 2020, *Nat. Commun.*) and is consistent with recent HERFD-XANES studies (Kvashnina et al., 2013, *Phys. Rev. Lett.*; Leinders et al., 2017, *Inorg. Chem.*), which firmly established the peak position of pentavalent uranium at the M_4 -edge. This clarification has been added to the Methods to ensure transparency in oxidation state assignments.

[Revised Methods, lines 599–602; Supplementary Information 2, lines 76–83]

- 7. The identification of U(IV) as the dominant oxidation state, with concurrent detection of U(V), adds important nuance to our understanding of microbial uranium reduction. This aspect of the study is a clear strength.**

We agree that the concurrent identification of U(IV) as the dominant oxidation state together with the persistence of U(V) is a key strength of our study. We have emphasised this aspect in the revised manuscript to highlight its significance for understanding microbial U reduction.

- 8. The manuscript notes a delay in uranium reduction. Please elaborate on whether this could result from microbial acclimatization, strong carbonate complexation inhibiting reduction, or other chemical/biological factors**

We agree that the initial delay in U reduction is an important observation. We interpret this delay as the result of a combination of microbial and geochemical factors, but our experimental design does not allow quantitative discrimination of these processes.

On the microbial side, the community likely required an acclimatization period to metabolize glycerol and generate intermediates such as acetate and H_2 among others, which serve as electron donors to stimulate U(VI)-reducing bacteria. This interpretation is supported by our monitoring data, which show a progressive decrease in sulphate and E_h following glycerol addition.

On the geochemical side, strong carbonate complexation of U(VI), together with the neutral to slightly alkaline pH of the mine water, likely reduced U bioavailability and slowed its reduction. In addition, competition with alternative electron acceptors such as, sulphate and Fe(III), may also have further contributed to this delay.

- 9. The assignment of U(V) is a key conclusion. Please clarify whether this spectral feature was confirmed through comparison with known U(V) reference spectra or solely inferred from literature.**

The assignment of U(V) is indeed central to our conclusions. We confirm that it was not based solely on literature values, but validated through direct comparison with reference spectra. U(IV) and U(VI) energy positions were calibrated with UO₂ and uranyl nitrate standards measured during the same beamtime, while the U(V) feature was benchmarked against the UMoO₅ reference spectrum reported by Pan et al. (Pan et al., 2020, *Nat. Commun.*) and found to be consistent with recent HERFD-XANES studies of mixed-valence oxides (Kvashnina et al., 2013, *Phys. Rev. Lett.*; Leinders et al., 2017, *Inorg. Chem.*).

In addition, UMoO₅ has been compared with other well-characterised U(V) compounds including NaUO₃, KUO₃ (Kvashnina and Butorin, 2022, *Chem. Commun.*) and U₄O₉ (Kvashnina et al., 2013, *Phys. Rev. Lett.*). These species exhibit identical HERFD-XANES peak positions and spectral shapes, which further supports the robustness of the U(V) assignment in our system.

[Revised Methods, lines 599–602; Supplementary Information 2, lines 76–83]

10. Given the well-known instability of U(V), a brief discussion on the potential stabilization mechanisms in your system—such as complexation with biogenic ligands or mineral surface association—would be informative.

We agree that the potential stabilization mechanisms of U(V) should be explicitly addressed. In the revised Results and Discussion, we now emphasise that U(V) persistence in our system is likely multifactorial. Contributing factors include complexation with biogenic ligands, incorporation into Fe-containing nanoparticulate phases such as FeU^(V)O₄, and adsorption onto microbial biomass or mineral surfaces. Together, these coupled biotic–abiotic processes provide a coherent and plausible explanation for the stability of U(V) under environmentally relevant conditions.

[Results and Discussion, lines 462–504]

11. Table S2 attributes uranium loss in control microcosms to biosorption. Please provide either references supporting this conclusion or additional data distinguishing biosorption from abiotic adsorption/reduction.

We recognise that the decrease of U observed in the control microcosms cannot be conclusively attributed to biosorption alone. Abiotic processes such as adsorption onto mineral particles or, potentially, container surfaces may also account for part of the removal. To reflect this uncertainty, we have revised the manuscript and now refer to this loss more conservatively as abiotic/adsorptive removal. This interpretation is consistent with previous studies showing that U can be removed through adsorption to mineral phases (Veeramani et al., 2013, *Environ. Sci. Technol.*; Pan et al., 2020, *Nat. Commun.*) or to microbial biomass (Lakaniemi et al., 2019, *J. Hazard. Mater.*; Fomina and Gadd, 2014, *Bioresour. Technol.*). This clarification does not alter the overall conclusions of the study, which are supported by the combined geochemical, microbiological, and spectroscopic evidence.

[Revised Results and Discussion, lines 144–147]

Response to Reviewers and Editor:

This document provides our detailed, point-by-point responses to the comments raised by the editor and reviewers on the manuscript “*Pentavalent and Tetravalent Uranium Formation via Glycerol-Stimulated Bacteria in Mine Water*” (NCOMMS-25-25237B).

We appreciate the editor’s and reviewers’ constructive comments, which have helped us to address key points and improve the overall quality of the manuscript.

As requested by the journal, all modifications made in the revised manuscript are indicated in **blue font** for ease of reference.

Reviewer #1 (Remarks to the Author):

1. The authors addressed my concerns and comments. I agree to accept the paper's publication.

We thank the Reviewer for their careful reading and constructive input, which helped improve the manuscript. We also appreciate their positive evaluation of the revised version.

Reviewer #2 (Remarks to the Author):

The revised manuscript is improved, but several issues still need further clarification.

1. The TEM data lack panoramic images that include multiple cells. Although the authors state that five cells and 231 nanoparticles were examined, the manuscript and SI present only close-up TEM/HRTEM images from single or very limited cells. Without panoramic views showing multiple cells within the same field of view, the claim of consistent nanoparticle formation across cells is not adequately supported.

To address this point, we have added panoramic TEM images showing multiple complete cells within the same field of view (Fig. S7), allowing us to evaluate the consistency of U-nanoparticle formation across different cells.

The panoramic images (Fig. S7 A–B) display different complete cells, whereas the Figure S7 (C–F) images correspond to the specific cells used for the quantitative analysis (Cells 1–5). These images provide detailed visualisation of the morphology and cellular localisation of the U-containing nanoparticle agglomerates. This confirms that the localisation patterns observed in the high-resolution analyses (Fig. 3 and Fig. 4) are not isolated cases but are representative of a larger number of cells.

We have also revised the main text to introduce this information explicitly in the Results section.

The revised text now reads:

“...To assess whether nanoparticle formation was consistent across cells, panoramic electron microscopy images were obtained and are shown in Figure S7. These images display multiple complete cells within the same field of view (Fig. S7 A–B) and reveal that the electron-dense agglomerates appear as bright features with a reproducible cellular distribution across independent cells (Fig. S7 C–F)...”

[Revised Results and Discussion lines 295–300 and Supplementary Material Fig. S7 line 138]

2. The use of uranium removal percentages (20%, 60%, 90%) to define reaction stages is conceptually misleading, as uranium reduction and the associated microbial and geochemical processes are inherently time-dependent. Reaction time should serve as the primary criterion for defining reaction stages.

We appreciate the reviewer's concern regarding the terminology used to describe the reaction process. Our initial use of U removal percentages was intended as an operational sampling criterion, aimed at selecting black precipitates formed under clearly distinct chemical and redox conditions, rather than defining kinetic reaction stages. In complex microbially mediated systems such as mine-water microcosms, reaction kinetics are non-linear and do not proceed through discrete phases. Therefore, the extent of U removal provides a practical descriptor of the system state at any given time.

Nevertheless, we agree that without explicit clarification this approach could be misinterpreted as defining "stages". Accordingly, in the revised manuscript we have reformulated the text to use incubation time as the primary reference, while using U removal percentages solely as operational descriptors of the system state at the time of sampling.

Specifically, we have updated the Results/Discussion, Methods, and figure legends (Fig. 1, Fig. S5, Fig. S13,) to explicitly link each sampled solid phase to the corresponding temporal window in the evolution of dissolved U concentration and redox potential (Fig. S2–S3). Based on the time-course data, the three sampled conditions correspond approximately to 10 days (day 10; ~20% decrease), 30 days (days 20–40; ~60% decrease), and 55 days (days 52–60; ~90% decrease) of incubation. We also explicitly state that these values characterise the chemical/redox state of the system at the time of analysis and should not be interpreted as discrete kinetic stages.

[Revised Results and Discussion lines 168–180; 184–191, 204–209, 301–303, 325–327; Methods lines 580–587, 591–592, 652–654; Supplementary Material lines Fig. S5 (lines 127–132) and Fig. S13 (lines 182–188)]

3. **The interpretation of the unusually low S_0^2 value (0.60) in the EXAFS fitting is not physically justified. S_0^2 should remain relatively constant for a given absorption edge. A refitting procedure with S_0^2 fixed to the reference value is recommended to test the robustness of the structural model.**

Raw U L_3 -edge spectra are not provided. Only processed HERFD-XANES and EXAFS data are presented, preventing independent evaluation of data quality and processing procedures. Raw spectral data should be included in the supplementary materials.

Minor Revision is therefore recommended before the manuscript can be further considered.

We thank Reviewer 2 for drawing our attention to the issue of the unphysically low S_0^2 . We used this comment to conduct a rigorous investigation into the origin of the low S_0^2 value.

After careful inspection of this analysis, we came to the conclusion that fixing S_0^2 to the reference value, as obtained for the U(V/VI) carbonato references ($S_0^2 = 0.96$), and fixing the CN of O_{ax} and O_{eq} at physically justified values gives more reliable results for the isolated spectrum of the biogenic U(V/VI) carbonato species.

Thus, we agree with the Reviewer's advice and have revised the manuscript accordingly.

We included the raw U L_3 -edge XAS spectra in the Supplementary Material.

For our investigation, we performed the following procedure:

Question: What is the origin of the unusually low S_0^2 obtained for the isolated U(V/VI) carbonato species?

Workflow:

1. Preparation of a theoretical example consisting of three spectral mixtures without and with experimental error simulated by using Gaussian random noise.
2. Isolation of the spectrum of the hypothetical U(V/VI) carbonato species by ITFA from the spectral mixtures
3. Shell fit of the isolated spectrum with different shell fit parameter settings

For step 1, three linear combinations (Fig. 1) were calculated using the spectrum of crystalline uraninite and the theoretical spectrum of a hypothetical U(V/VI) carbonato species (Fig. 2), while the fractions were taken from the uraninite-like and the U(V/VI) carbonato species (Fig. 3). The spectrum of the shell fit of the U(V/VI) carbonato species (Tab. 1 and Fig. 2 in the manuscript) serves as the spectrum of the hypothetical U(V/VI) carbonato species for which the EXAFS structural parameters are given in Table 1.

The choice of this spectrum is supported by the expectation of the presence of structurally slightly different U(V/VI) carbonato species which causes relatively high DW factors ($DW_{O_{ax}} = 0.013 \text{ \AA}^2$, $DW_{O_{eq}} = 0.008 \text{ \AA}^2$, $DW_C = 0.014 \text{ \AA}^2$).

Table 1. EXAFS structural parameters of the hypothetical U(V/VI) carbonato species (taken from Table 1 in the manuscript without round-off)

Path	CN	R/Å	DW/Å ²	dE0/eV	Fit number R-value/%
Hypothetical U(V/VI) carbonato species, $S_0^2 = 0.96$					
O _{ax}	2	1.8467	0.012570	7.1746	/
MS U-O _{ax(1)} - O _{ax(2)}	2	3.6934/	0.025140	7.1746	
O _{eq}	5	2.4563	0.008372	7.1746	
C	2.5	2.9257	0.013617	7.1746	

Fig. 1. Three spectral mixtures calculated using the spectra of crystalline uraninite and the hypothetical U(V/VI) species (see Fig. 2) and the fractions observed for the uraninite-like and the U(V/VI) carbonato species (see Fig 3), respectively. The numbers correspond to the spectrum number in Fig. 3. Gaussian noise was added to the mixtures.

Fig. 2. Spectra of the crystalline uraninite (red) and the hypothetical U(V/VI) carbonato species (blue) used for the calculation of the three spectral mixtures in Fig. 1. ITFA isolated spectra of the crystalline uraninite and the hypothetical U(V/VI) carbonato species (both black).

Fig. 3. Fractions of the uraninite-like spectrum (red) and the U(V/VI) carbonato species (blue) used for the calculation of the three spectral mixtures in Fig. 1.

With respect to the added noise the ITFA isolated spectra agree very well with the spectra of crystalline uraninite and the hypothetical U(V/VI) carbonato species (Fig. 2), while the U-U interaction at $3.64 + \Delta\text{\AA}$ is completely removed in the case of the isolated spectrum of the

hypothetical U(V/VI) carbonato species. If no noise is added the ITFA-isolated spectra agree perfectly with the spectra introduced for the calculation of the spectral mixtures.

We performed several trials of shell fit parameter settings in order to fit the isolated spectrum of the hypothetical U(V/VI) carbonato species (Tab. 2), while the corresponding fits are shown in Fig. 4. If no noise is added to the spectral mixtures the shell fit agrees within the numerical uncertainty with the introduced structural parameters of the model (compare Tab. 1 with Tab. 2 Fit number 1), hence the ITFA isolation procedure works – zero test passed.

In the presence of noise and keeping $S_0^2 = 0.96$ the CN and DW differ in partly strongly from the theoretical model values (Tab. 1) and show the usual positive correlation effects (Tab. 2 Fit number 2). The deviations of the interatomic distances from the model values are much less pronounced and are near at the common error of 0.02 \AA in their estimation.

If we use the parameter setting of our previous submission (Tab. 1 in the manuscript) then we achieve a similar physically too small $S_0^2 = 0.67$ (vs. 0.60 in the manuscript), hence we can reproduce this shortcoming of the fit (Tab. 2 Fit number 3). The increase of the CN_{Oeq} to an unphysical value of 8 (vs. 7 in the manuscript) could also be reproduced. However, again the deviations of the interatomic distances from the model values are much less pronounced and close to the common error in their estimation of 0.02 \AA .

Finally, we fixed CN_{Oax} and CN_{Oeq} to physically realistic values of 2 and 5 , respectively, and set $S_0^2 = 0.96$ (Tab. 2, Fit number 4). This constrained fit showed the best match of the free structural parameters with those of the model (Tab. 1).

Table 2. Shell fit trials of the ITFA isolated spectrum of the hypothetical U(V/VI) carbonate species.

Path	CN	R/Å	DW/Å ²	dE0/eV	Fit number R-value/%
No noise added to the spectral mixtures, $S_0^2 = 0.96$					1 2.9*10 ⁻⁶
O _{ax}	2.0000(2)	1.846708(5)	0.012572(1)	7.1755(4)	
MS U-O _{ax(1)} - O _{ax(2)}	2.0000/	3.693415/	0.025144/	7.1755/	
O _{eq}	4.9999(4)	2.456295(3)	0.008372(1)	7.1755/	
C	2.5000/	2.92568(2)	0.013614(2)	7.1755/	
Noise added to the spectral mixtures, $S_0^2 = 0.96$					2 6.39
O _{ax}	1.4(2)	1.833(5)	0.008(1)	5.5(5)	
MS U-O _{ax(1)} - O _{ax(2)}	1.4/	3.666/	0.016/	5.5/	
O _{eq}	5.5(5)	2.443(4)	0.0080(9)	5.5/	
C	2.7/	2.89(3)	0.016(4)	5.5/	
Noise added to the spectral mixtures, adjusting S_0^2 to get CN _{Oax} = 2, gives: $S_0^2 = 0.67$					3 6.39
O _{ax}	2.0(3)	1.833(5)	0.008(1)	5.6(5)	
MS U-O _{ax(1)} - O _{ax(2)}	2.0/	3.667/	0.016/	5.6/	
O _{eq}	7.9(7)	2.443(4)	0.008(9)	5.6/	
C	3.9/	2.89(3)	0.016(4)	5.6/	
Noise added to the spectral mixtures, CN _{Oax} and CN _{Oeq} were set to physically realistic values and fixed, $S_0^2 = 0.96$					4 6.75
O _{ax}	2.f	1.840(6)	0.0122(6)	6.5(4)	
MS U-O _{ax(1)} - O _{ax(2)}	2./	3.680/	0.0245/	6.5/	
O _{eq}	5.f	2.448(4)	0.0073(3)	6.5/	
C	2.5/	2.91(2)	0.016(4)	6.5/	

CN = coordination number, R = radial distance, DW = Debye-Waller factor, dE0 = shift in energy threshold, S_0^2 = amplitude reduction factor, f = fixed parameter, / = linked parameter. MS = multiple scattering path along the uranyl chain. Estimated standard deviations of the variable parameter in parenthesis. Fitted k-range 3.00 Å⁻¹ – 11.95 Å⁻¹. R value calculated with:

$$R = \frac{\sum_i [\Delta(k^n \chi_{\text{exp}}(k_i)) - \Delta(k^n \chi_{\text{calc}}(k_i))]^2}{\sum_i (k^n \chi_{\text{exp}}(k_i))^2} \times 100\%$$

Fig. 4. Shell fit trials of the isolated spectrum of the hypothetical U(V/VI) carbonato species. Numbers correspond to the fit numbers in Tab. 2.

The investigation showed that experimental errors intrinsically lead to significant parameter correlations, which must be minimized to obtain a physically realistic solution. Furthermore, the interatomic distances are much less affected by experimental error and should therefore be used for drawing conclusions rather than other structural parameters, as already noticed in the manuscript. However, constraining $CN_{O_{ax}}$ and $CN_{O_{eq}}$ to physically meaningful values and setting S_0^2 to a value observed for the matching references (here $S_0^2 = 0.96$) will yield the most reliable EXAFS structural parameters. Thus, we agree with the referee's constructive suggestion and have revised the corresponding parts of the manuscript accordingly.

[Revised Results and Discussion lines 215–293; Methods lines 634–635; Supplementary Material lines Fig. S12 (lines 175–180)]

Reviewer #3 (Remarks to the Author):

The authors have revised the manuscript accordingly. I recommend publication after addressing the following minor concerns.

- 1. Although the potential indirect abiotic reduction of U by biogenic hydrogen sulfide and/or Fe(II) cannot be ruled out, the authors did not measure hydrogen sulfide or Fe(II) species in their samples. Caution should therefore be exercised when discussing this process in the Conclusion. The Conclusion should also be more succinct and focused on the major findings.**

We agree that, because H₂S and Fe(II) were not quantified in our incubations, indirect abiotic U(VI) reduction involving these species cannot be confirmed. In response, we have revised the relevant passage in the Conclusion to ensure that these pathways are presented as plausible but not demonstrated, avoiding any wording that could imply direct evidence. The revised text now reads:

L449–451: “...*Together with the observed decrease in sulphate content in the microcosms, our data indicate that Desulfobulbus and Desulfovibrio reduced sulphate to H₂S, which could potentially contribute to indirect abiotic U(VI)...*”

L453–455: “...*As a result, U(VI) reduction in the microcosms likely involved a combination of direct enzymatic pathways and indirect abiotic contributions mediated by biogenic H₂S...*”

L540–543: “...*These bacteria contribute directly to U(VI) reduction and help maintain reducing conditions, while indirect abiotic contributions involving biogenic sulphide or Fe(II) remain plausible but cannot be assessed with our dataset...*”

[Revised Results and Discussion lines 449–451, 453–455; Conclusions lines 540–543]

As recommended, we also streamlined the Conclusion section to sharpen its focus on the major outcomes of the study, and the implications of these findings for U mobility and bioremediation. These revisions shorten the section and improve its clarity and precision.

[Revised Conclusions lines 526–560]

- 2. L518–521, While fermenters can generate electron donors from glycerol to support SRB activity, the authors previously noted that SRB can utilize glycerol directly (L410–413). This point should be clarified for consistency.**

The two statements refer to complementary processes that operate simultaneously in our system. As noted in the manuscript, *Desulfobulbus* and *Desulfovibrio* are well documented to utilise glycerol directly, while fermenters degrade glycerol into intermediates (e.g., acetate, lactate, H₂) that support SRB activity. To ensure consistency between these statements, we have revised the text to explicitly reflect this dual contribution. The updated text now reads:

“...*Fermenters generate electron donors from glycerol degradation that promote SRB activity. However, many SRB are also known to use glycerol directly...*”

[Revised Conclusions lines 538–540]

- 3. L48–49, L669–671, grammatically incorrect.**

We agree with the reviewer’s comment. The two sentences have been revised to correct the grammatical issues and to improve clarity, without altering their scientific meaning. The updated versions now read:

L48–49: “...*Elevated U concentrations in surface and groundwater frequently exceed the drinking-water guideline of 0.03 mg/L⁴....*”

[Revised Introduction lines 48–49]

L691–695: “...*To identify active metabolic processes within the mine-water community via differential gene expression (DGE), metatranscriptome profiling was also conducted on Pöhla mine water. This site is a former U-mining location in Saxony where the U concentration was 0.01 mg/L, approximately 100 times lower than in Schlema-Alberoda^{3,5}...*”

[Revised Methods lines 691–695]

- 4. U(IV) at L123; “hexavalent U” “Tetravalent U” at L167. The U(IV), U(V) and U(VI) should be spelled out in full at their first appearance in the manuscript. Thereafter, the abbreviated forms may be used.**

We have implemented the requested corrections. The first occurrences of each uranium oxidation state are now written out in full, and the abbreviated forms are used throughout the manuscript.

[Revised Introduction lines 59 and 61]

5. Fig. S7 is not cited in the manuscript.

The former Supplementary Figure S7 (now Figure S13), was previously cited in SI-2, where the ITFA procedure is described. To ensure that its role is clearly integrated into the manuscript, we now cite Supplementary Figure S13 in the Methods section and explicitly indicate the comparison it provides. The revised text reads:

“...The corresponding comparison between the experimental U M₄-edge HERFD-XANES spectra and the ITFA-derived model fits is shown in Figure S13...”

All supplementary figures are now explicitly cited in the main text.

[Revised Methods lines 643–645]

Reviewer #4 (Remarks to the Author):

1. The manuscript has been well revised and can be accepted for publication at current form.

We appreciate the reviewer’s thorough review and positive recommendation. The reviewer’s comments have helped us improve the clarity and quality of the manuscript.

Response to Reviewers and Editor:

This document contains our point-by-point responses to the comments from the editor and reviewers regarding the manuscript “*Pentavalent and Tetravalent Uranium Formation via Glycerol-Stimulated Bacteria in Mine Water*” (NCOMMS-25-25237C).

We are grateful for the editor’s continued guidance and the reviewers’ thoughtful feedback, which have helped us further clarify the scope and implications of our work. As per journal guidelines, all changes made in the revised manuscript are highlighted in **blue font** for clarity.

Reviewer #2 (Remarks to the Author):

The authors have carefully revised the manuscript. It is recommended for publication after the following minor revisions:

- 1. The manuscript suggests that mineral surface processes and Fe phases may contribute to U stabilization. Notably, beyond surface sorption/co-precipitation, U may also migrate into Fe-bearing clay interlayers and become persistently sequestered within confined microenvironments (doi: 10.1016/j.watres.2025.123582; 10.1016/j.gca.2009.07.002). The Discussion would benefit from a more detailed comparison of surface-driven mechanisms versus “interlayer confinement” processes, including their key distinctions and potential coupling, to better articulate the environmental relevance and practical implications of this study.**

We thank the reviewer for this comment and agree that an explicit comparison between surface-dominated immobilisation mechanisms and those associated with interlayer confinement in Fe-rich clays adds important conceptual clarity and strengthens the environmental relevance of the study.

In response, we have expanded the Discussion to contrast (i) interlayer confinement mechanisms reported for uranium in Fe-rich clays, including internal retention and redox transformations mediated by structural Fe, with (ii) the microbiological driven mechanism observed in our system, where reduced U species (U(IV) and U(V)) are stabilised at microbe–mineral interfaces and within biogenic Fe–U phases. We also clarify that structurally confined U(V) within clay interlayers has not yet been conclusively demonstrated.

Finally, we explicitly state that our study does not directly investigate clay minerals or interlayer processes, but instead reveals a distinct immobilisation mechanism operating under mine-water conditions. While these mechanisms are not mutually exclusive and may coexist in heterogeneous natural systems, they are governed by different redox, structural, and mineralogical controls, with distinct implications for the long-term stability of uranium.

[Revised Results and Discussion, lines 510–529]

Reviewer #3 (Remarks to the Author):

- 1. The manuscript has been revised accordingly and can be accepted for publication.**

We appreciate the reviewer’s thorough review and positive recommendation. Their comments have helped improve the clarity and overall quality of the manuscript.